# Whole body regeneration deploys a rewired embryonic gene regulatory network logic

Rita Andreoni-Pham [1,4], Hereroa Johnston [1,2,4], Jacob F. Warner [1,3,4], Karine Nedoncelle[1], João E. Carvalho [1], Kai Hofmänner [1], Agata Maugeri[1], Olivier Croce [1], Aldine R. Amiel [1,5] ✉ & Eric Röttinger [1,5] ✉

A long-held hypothesis in regeneration proposes that developmental processes are re-deployed during regeneration. To investigate this, we compared embryonic and regeneration gene regulatory networks (GRN) in the sea anemone *Nematostella vectensis* using transcriptomic time series spanning these two developmental trajectories. Here, we show that regeneration reuses cohorts of the embryonic genes along with a small set of genes whose expression dynamics are specific to regeneration. We identified co-expression modules that are either conserved between embryogenesis and regeneration or specific to regeneration, with the latter linked to cellular mechanisms such as apoptosis, tissue remodeling, and wound healing. Functional assays revealed that apoptosis and cWnt signaling pathways are partially MEK/ERK dependent, have largely distinct downstream targets but converge to coordinate regenerative responses. Collectively, these results indicate that regeneration in *N. vectensis* represents a partial redeployment and extensive rewiring of the embryonic GRN, reactivating developmental modules through a regeneration-specific network logic.

Regeneration of cells, tissues, appendages, or even entire body parts is a widespread yet still a rather poorly understood phenomenon in the animal kingdom[1]. A long-standing question in the field of regeneration is whether and to what extent embryonic gene programs, initially used to build an organism, are reused during regeneration[2,3]. Several transcriptomic studies of regeneration in axolotl, anole, zebrafish, and sea anemones have highlighted the importance of the re-deployment of developmental pathways[4–10]. Many studies have directly compared embryonic and regenerative gene expression, focusing on single or groups of genes, revealing (i) genes that are specific to embryonic development[11], (ii) genes that are specifically expressed or required during regeneration[12,13], and (iii) embryonic genes that are reused during regeneration to some extent[14–18].

One study has addressed the question by comparing the transcriptional dynamics between post-larval development and regeneration from dissociated cells in sponges, further highlighting partial similarities between these two developmental trajectories[19]. Another study has investigated the relationship between larval skeleton development and brittle star arm regeneration with a special emphasis on FGF signaling, revealing that a skeleton developmental gene regulatory module is re-deployed during regeneration[20]. Investigating the morphological, cellular, and transcriptomic relationship between leg development and regeneration in a marine arthropod (*P. hawaiiensis*), the authors described (i) that regeneration produces an exact replica of the leg[21] and (ii) that both processes involve similar sets of genes, although their temporal relationship seems to be different[22].

Despite these studies, the extent to which gene regulatory network programs are reused globally during regeneration, as well as the functional comparison between these two developmental trajectories, remains unknown. The sea anemone *Nematostella vectensis* (Cnidaria,

[1]CNRS, INSERM, Institute for Research on Cancer and Aging, Nice (IRCAN), Université Côte d'Azur, Nice, France. [2]Present address: Tahiti Marine Biotech, Cote Mer - Fare Ute Port De Pêche Local N°5, Papeete, French Polynesia. [3]Present address: Department of Biology and Marine Biology, University of North Carolina Wilmington, Wilmington, NC, USA. [4]These authors contributed equally: Rita Andreoni-Pham, Hereroa Johnston, Jacob F. Warner. [5]These authors jointly supervised this work: Aldine R. Amiel, Eric Röttinger. ✉e-mail: aldine.amiel@univ-cotedazur.fr; eric.rottinger@univ-cotedazur.fr

Anthozoa) offers a unique lens into this question as it affords large-scale functional comparisons between embryonic development and whole-body regeneration within the same organism.

*Nematostella vectensis* (*N. vectenis*) is a research model in which embryonic development and whole-body regeneration can be studied (Fig. 1A), and thus, ideally suited for this line of inquiry[23]. *N. vectensis* has long been used as a model system to study embryonic development, evolution of body patterning, and gene regulatory networks[24–30]. More recently, *N. vectensis* has emerged as a powerful whole-body regeneration model as it is capable of re-growing missing body parts in less than a week[9,31–39], reviewed in ref. 40.

Local injury-induced regeneration in *N. vectensis* causes a systemic response throughout the entire body that is crucial for maintaining shape homeostasis during regeneration[41]. This process follows a

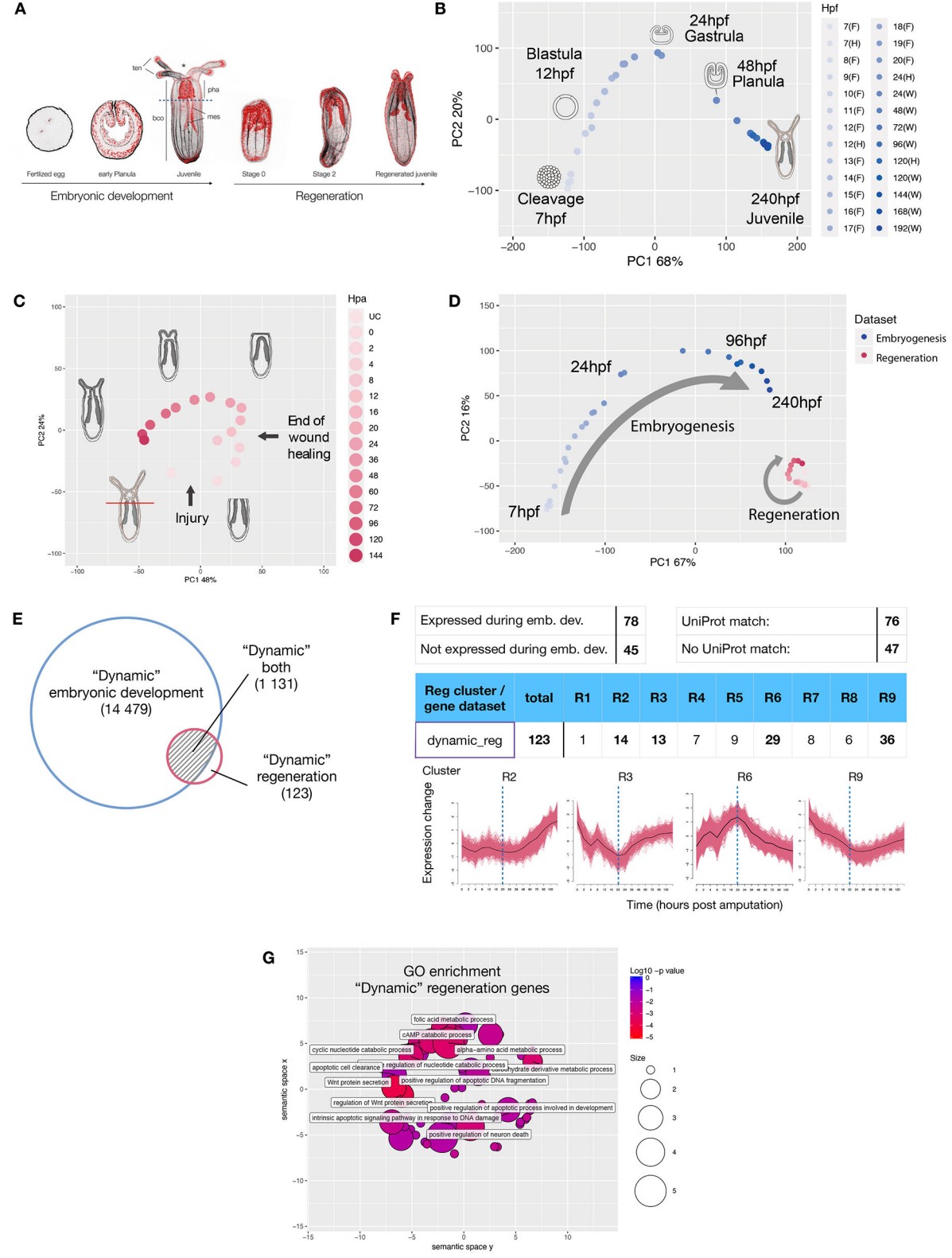

**Fig. 1 | Comparison of embryonic and regenerative transcriptomes. A** General morphology of *N. vectensis* during embryonic development and regeneration (black; f-actin/Phalloidin, red; nuclei/DAPI). Dashed line: amputation site, ten tentacles, pha pharynx, mes mesenteries, bco body column. **B** Principal component analysis (PCA) of three batch-corrected embryonic datasets[39,43,44]. Plot legend indicates timepoint and dataset (Fischer et al.: Helm, F et al.:(H) and Warner et al.:(W). Most of the variation is observed in the first 24 h of development. **C** PCA of regeneration dataset sampled at uncut (UC), 0, 2, 4, 8, 12, 16, 20, 24, 36, 48, 60, 72, 96, 120, and 144 hpa. Data were extracted from the NvERTx database[39]. Regeneration proceeds through a wound-healing phase (0–4 hpa) followed by the early regenerative program (8–20 hpa) and ending with a late regenerative program which approaches the uncut condition (24–144 hpa). **D** PCA of embryonic versus regeneration samples. Embryogenesis (blue) exhibits far greater transcriptomic variation than regeneration (red). **E** Comparison of differentially expressed (|log2(FC)| > 2 and FDR < 0.05 for any timepoint comparison against $t_0$, where $t_0$ = 7 hpf for embryogenesis and 0 hpa for regeneration) "dynamic" genes during embryogenesis (blue) and regeneration (green). **F** Global overview of the regeneration specific genes expression and classification, referring to clusters shown in Fig. S2. **G** GO term enrichment for dynamic regeneration genes. GO-term enrichment was assessed using a one-sided Fisher's exact test ($P < 0.02$).

dynamic but highly stereotypical morphological and cellular program (i.e., regeneration stages), involving tissue remodeling and the de novo formation of body structures[35,36]. More precisely, regeneration requires a Tissue Contact (*TC*) between the mesenteries (*MES*) and the body wall epithelia (*BWE*) at the amputation site[37], cell proliferation[34,36], as well as two populations of fast and slow cycling, potential stem cells[37]. It has been suggested that MEK/ERK signaling is required for oral regeneration[38], and recently, a study has shown that ROS (Reactive Oxygen Species) signaling is crucial for post-*TC* regeneration steps by controlling cell proliferation in a tissue-specific manner[42]. While many developmental signaling pathways are deployed during regeneration[9,38,39,41,42], their precise roles as well as their regulatory logic remain unknown.

Here, we take advantage of the biological and experimental properties of *N. vectensis* to address the historical question to what extent regeneration recapitulates embryonic development and to decipher molecular signatures unique to regeneration. In the present study, we performed a global transcriptomic comparison of embryogenesis and regeneration using deeply sampled transcriptomic datasets. This approach was combined with a pathway-specific functional analysis as well as a comparison of the embryonic and regeneration GRNs within the same species. Overall, this study (i) revealed that whole-body regeneration is transcriptionally modest in comparison to embryonic development; (ii) identified genes, cellular processes and network modules specific to the process of regeneration; (iii) demonstrated the plasticity of the network architecture underlying two developmental trajectories; and finally, (iv) showed that whole-body regeneration deploys a rewired embryonic GRN logic to reform lost body parts.

## Results
### Regeneration is a partial re-use of embryonic development
To compare embryogenesis and regeneration (Fig. 1A) on a global transcriptome-wide scale we analyzed four RNAseq datasets; one spanning 16 time points of regeneration[39] and three spanning a total of 29 embryonic time points[39,43,44] that were batch corrected[39]. To assess the transcriptomic states underlying embryogenesis we performed principal component analysis (PCA) on batch corrected embryonic data (Fig. 1B). We found that most gene expression changes occur during the first day of embryonic development from cleavage to blastula stage (Fig. 1B, 7 h post fertilization (hpf)–24 hpf, PC1 proportion of variance 68%; PC2 proportion of variance 20%) indicating large transcriptomic differences in early embryogenesis. From 96 hpf onwards the samples exhibited modest changes in transcriptional variation suggesting that most transcriptional dynamics driving embryogenesis is complete by this stage (96–240 hpf).

When we examined the regenerative program using PCA (Fig. 1C), we observe three distinct transcriptional programs based on the variations in gene expression: (i) a wound-healing phase (0–4 h post amputation (hpa)) that is followed by (ii) the activation of the early regenerative program (8–20 hpa) in which the samples are distributed along the second principal component (PC2 proportion of variance = 24%, Fig. 1C). These two phases can be correlated to morphological

and cellular processes observed in earlier studies, i.e., the completion of wound-healing at 6 hpa[36] as well as the onset of mitotic activity 12–18 hpa[34,36]. (iii) From 24 hpa onwards, most of the variation in gene expression is explained by the first principal component during the late regenerative phase (PC1 proportion of variance = 48%, 24–144 hpa, Fig. 1C). This phase corresponds to the burst of cell proliferation as well as the reformation of oral structures such as the pharynx, the tentacles and the mouth[36] (iv) Towards the end of regeneration (144 hpa), we observe a transcriptomic profile that resembles the uncut samples indicating a return to steady state. These profiles correlate with the major events of oral regeneration in *N. vectensis* and indicate that our sampling strategy effectively covers the major transcriptional hallmarks of regeneration[36].

We next directly compared the transcriptomic variation of regeneration and embryogenesis using PCA and found that the transcriptional changes during regeneration were relatively modest compared to those observed during embryogenesis with most of the variation in the first two principal components being driven by the embryonic data (PC1 proportion of variance = 67%, PC2 proportion of variance = 16%, Fig. 1D). This indicates that the transcriptional dynamics of embryogenesis are more profound than those of regeneration. This finding was buttressed by comparing the number of 'dynamically expressed genes', those which are significantly differentially expressed $\log_2$FC > 1 or < −1 and false discovery rate (FDR) < 0.05 at any time point compared to $t_0$ defined as 0 hpa for regeneration and 7 hpf (the onset of zygotic transcription) for embryogenesis. Embryogenesis exhibited more than ten times the number of dynamically expressed genes compared to regeneration (15,610 and 1254 genes, respectively, Fig. 1E). These results further show that regeneration, when compared to embryogenesis, employs far fewer dynamic genes to accomplish a similar task: (re)constructing a functional animal. Of the dynamic genes observed during regeneration, the vast majority (1131 out of 1254) are also dynamically expressed during embryogenesis, suggesting that regeneration is largely a partial re-use of the embryonic gene complement (Fig. 1E).

### Identification of genes with regeneration-specific expression dynamics
Among the genes dynamically expressed during regeneration, a small fraction, 123 genes, exhibit significant differential expression (fC > 2-fold) only during regeneration which we term "regeneration specific" (Supplementary Data 1). There is no obvious bias into a specific temporal expression pattern, however, they are enriched in the regeneration clusters[39] R2, R3, R6, and R9 (Fig. 1F). Furthermore, 45 of these genes are detectable exclusively during the regeneration process indicating they are transcriptionally silent during embryogenesis. The remaining 78 genes are expressed during embryonic development but are not considered dynamic from our differential expression analysis above (Fig. 1F). Interestingly, several of the 123 "regeneration specific" genes, for example *wntless* (NvERTx.4.10017) and *agrin-like* (NvERTx.4.43369), have previously been reported to be important regulators of regeneration in bilaterians[45,46]. Furthermore, among the "regeneration specific" genes, 47 have no known homology in the

Uniprot database (BLASTp, *e* value cutoff <0.05, see "Methods" for annotation details), suggesting possible taxonomically restricted genes. These results indicate not only a possible evolutionary conservation of gene use in regeneration but also identify additional genes that may play important roles specifically during whole-body regeneration. A gene ontology (GO) term enrichment analysis on these 123 "regeneration specific" genes, revealed a suite of biological process GO terms relating to modulation of signaling pathways (e.g., *wntless*), metabolic processes, and apoptotic cell death, indicating an essential role for these processes in regeneration (Fig. 1G).

## Embryonic gene modules are partially redeployed during regeneration

While keeping in mind the difference between the two processes, i.e., development of an entire organism vs regeneration of oral structures, it is nonetheless interesting to note that regeneration dynamically expresses less than one-tenth the number of genes compared to embryonic development. We were thus interested in how these genes were deployed and arranged into co-expression networks. We sought to determine if embryonic gene network modules themselves are reused in a reduced capacity or if regeneration deploys novel gene module arrangements. For this, we used previously determined fuzzy c-means gene expression clusters[39], i.e., eight embryonic clusters (Figs. S1 and S2A) and nine regeneration clusters (Figs. S1 and S2B).

To explore these expression clusters (also named modules), we performed GO-term enrichment for each cluster (Supplementary Data 2 and 3) and examined the clusters at the gene level. We found that modules that were activated early in either of the two processes (e.g., embryogenesis cluster 4 (E4), regeneration cluster 6 (R6), Fig. S2A, B) contained several canonical developmental genes: *wntA* (NvERTx.4.132141), *lmx* (NvERTx.4.163477) and *foxA* (NvERTx.4.73096) in embryonic cluster 4 (Fig. S2Ai) and *tcf* (NvERTx.4.100051), *spr* (NvERTx.4.64399), and *runx* (NvERTx.4.92298) in R6 (Fig. S2Bi). The early embryonic transcriptional activation of *wntA, lmx and foxA* in the endomesoderm/central ring has been previously reported[26,47–49] and confirmed by in situ hybridization at 24 hpf (Fig. S2Aii). The spatio-temporal expression patterns of *tcf*, *spr* and *runx* during regeneration (uncut, 0–60 hpa, Fig. S2Bii) confirm the dynamic expression patterns of these genes as early as 2 hpa at the amputation site. We conclude from this analysis that classical developmental genes are involved in the early phases of both embryogenesis and regeneration.

We next analyzed whether the same groups of genes were co-regulated during embryogenesis and regeneration, by testing if gene expression observed during both processes was arranged in similar co-expression modules (Fig. 2). We compared regeneration and embryonic clusters on a gene-cluster membership basis to identify significant overlaps. Regeneration clusters with high overlap of a specific embryonic cluster indicate a shared or reused network logic since the same suite of genes is deployed as a bloc in both processes. Regeneration clusters with low overlap to any single embryonic cluster, on the other hand, are likely to be de novo genetic arrangements specific to regeneration.

We found that most of the regeneration clusters exhibited significant overlap with one or more embryonic clusters (Fig. 2A, B). These "conserved modules" also exhibited high preservation permutation co-clustering *z*-scores (Fig. 2B; >2 indicating conservation; >10 indicating high conservation; permutations = 1000)[50]. Importantly, we identified two clusters, R1 and R6, which exhibited relatively low overlap with any one embryonic cluster, indicating that these are likely "regeneration specific" arrangements. When we examined the GO-term enrichment of each cluster, we found that in general, highly conserved clusters (e.g., R5) were enriched in GO-terms corresponding to homeostatic cell processes (Fig. 2C), while lowly conserved regeneration specific clusters (e.g., R6) were enriched in GO-terms describing developmental signaling pathways (Fig. 2F). These results suggest

that the gene module arrangements pertaining to core biological functions such as cell proliferation are common to embryogenesis and regeneration while the gene modules containing developmental patterning genes are unique to each process.

Two clusters that exemplify these findings are R5 and R6. Cluster R5, a conserved cluster (zStatistic 6.94) showed strong enrichment of cell-proliferation related GO-terms (Fig. 2C). When we examined exemplar genes (with intra-module membership scores >0.95) *ercc6-like* (NvERTx.4.116232), *rad54B* (NvERTx.4.92456), *mcm10* (NvERTx.4.112709), *cyclinB3* (NvERTx.4.144958), we observed co-expression patterns during embryonic development (cluster E-1, Fig. 2D) and regeneration (cluster R5, Fig. 2E) further demonstrating module conservation. The expression pattern of these genes during regeneration correlate well to the timing of proliferation activation during *N. vectensis* regeneration with an activation at 24 hpa, a peak at 48 hpa, and a taper off thereafter[34,36] (Fig. 2E).

In contrast, cluster R6 is specific to regeneration. This module exhibits strong enrichment of GO-terms relating to apoptosis and developmental signaling pathways (Fig. 2F). When we examined four exemplar genes *tcf* (NvERTx.4.100051), *bax* (NvERTx.4.118493), *runx* (NvERTx.4.92298), *bcl2* (NvERTx.4.81296), we observed divergent profiles during embryogenesis (Fig. 2G) but co-expression during regeneration (Fig. 2H) indicating that this grouping of genes is indeed "regeneration specific". These results suggest that modules containing genes responsible for basic cellular functions are largely reused and co-expressed between embryogenesis and regeneration, while those including genes that are important for the activation of developmental processes have regeneration-specific arrangements.

## Identification of synexpression groups at the onset of morphogenetic movements

We next decided to characterize the potential synexpression groups comprising the GRN underlying whole-body regeneration in *N. vectensis* at 20 hpa. This time point was chosen as it corresponds to the onset of morphogenetic movements, invagination/tissue reorganization[36], as well as the likely onset of the regenerative program (Fig. 1C). Thus, this time point will allow a comparison with the previously defined embryonic GRN at the onset of the morphogenetic movements of gastrulation[26,51].

To identify a set of genes relevant for our study and potentially involved in coordinating the regenerative response we performed a differential gene expression analysis between 0 and 20 hpa and identified 2263 transcripts of which ~1400 were differentially expressed with an at least 2-fold variation (Fig. 3A; Supplementary Data 4). We cross-referenced this list of transcripts with the "regeneration-specific" genes identified above (Supplementary Data 1) as well as with genes that are part of the GRN underlying embryonic development, i.e., 15 embryonic cWnt targets[26] (Supplementary Data 5) and 25 embryonic MEK/ERK targets[51] (Supplementary Data 6) downstream targets. These embryonic downstream targets are largely pathway-specific as only 4 genes (*nfix-like*, *hd050*, *elk-like1*, and *gsc*) are controlled by MEK/ERK and cWnt at the onset of gastrulation (Supplementary Data 5 and 6). Doing so, we identified 10 embryonic GRN (6 MEK/ERK + 4 cWnt downstream targets) and 44 "regeneration-specific" genes that were downregulated at 20 hpa (Fig. S3; Supplementary Data 7,8, and 9). Furthermore, we identified 30 genes that are part of the embryonic GRN (19 MEK/ERK[51,52] and 11 cWnt[26] downstream targets) as well as 40 "regeneration-specific" genes that were upregulated at the onset of regeneration (Fig. 3B; Supplementary Data 7,8, and 9).

Performing an in situ hybridization screen for these genes we obtained 26 patterns that we regrouped based on their spatial expression profiles at 20 hpa (Fig. 3C). Genes were either expressed in the amputation site ectodermis (AE: *wnt2, foxB, wntA*; Fig. 3Ca–c), amputation site gastrodermis (AG: *wnt7b, nkd1-like, wnt4, fz10*; Fig. 3Cd–g), amputation site gastrodermis + amputation site

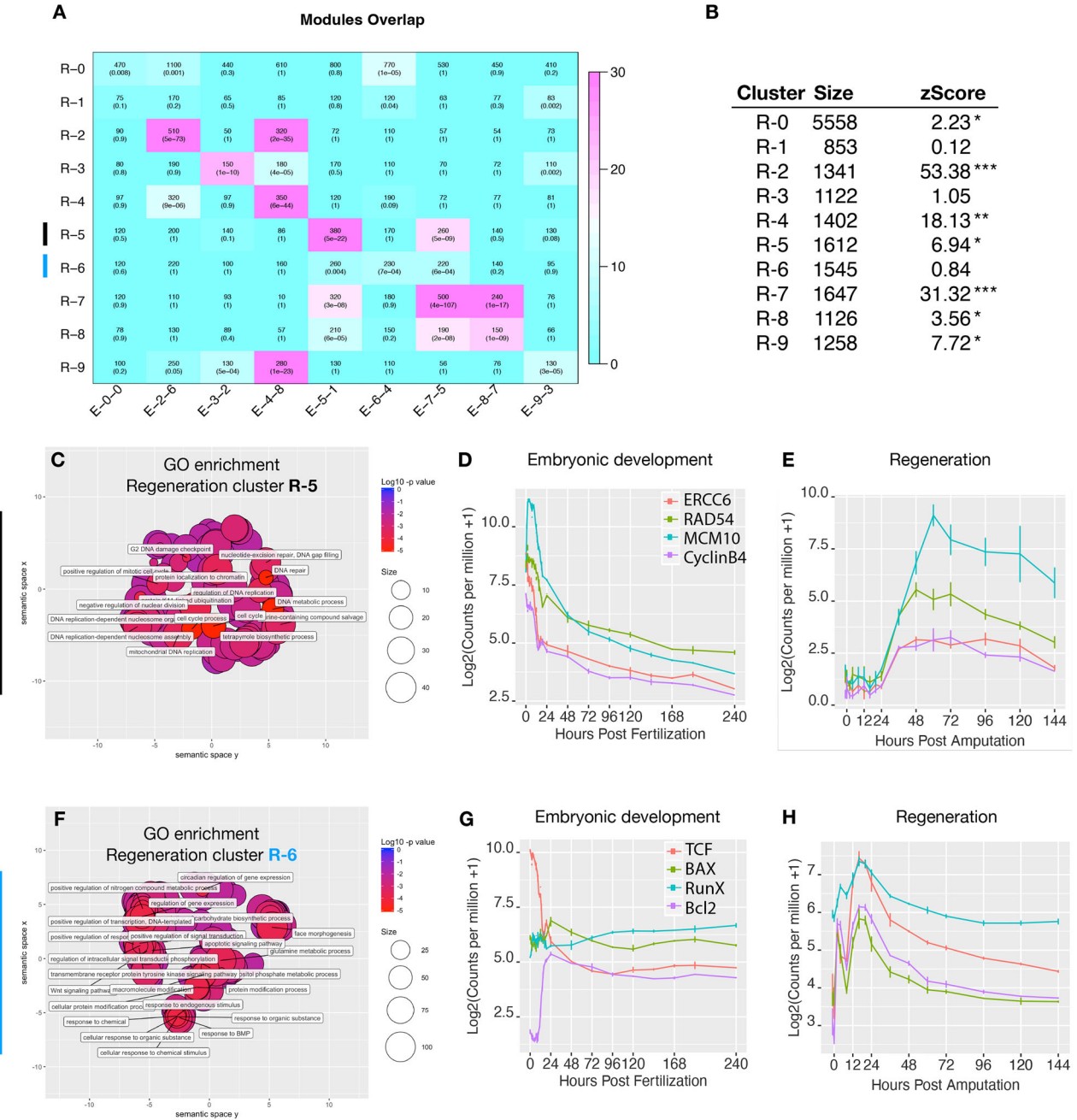

**Fig. 2 | Embryonic gene modules are partially redeployed during regeneration.** Overlap table of regeneration versus embryonic modules (**A**). In each cell, the overlap itself is quantified along with the p-value (Fisher's exact test). Color indicates –log10(pvalue), with a brighter magenta indicating a more significant overlap. R0 and E0 contain genes that are not assigned to any module. **B** Table indicating the size, and the co-clustering zStatistic. A zStatistic >2 (*) indicates moderate module conservation, >10 (**) high conservation >30 (***) very high conservation. **C** The conserved module R5, enriched in cell proliferation GO terms, with exemplar genes (*ercc6, rad54B, mcm10*, and *cyclinB3*) showing co-expression during embryonic development. Statistical analysis was performed using the regeneration cluster assignments as the reference set and 1000 permutation. **D** and regeneration (**E**). **F** The regeneration specific module R6, enriched in GO-terms associated with apoptosis and WNT signaling, with exemplar genes (*tcf, bax, runx*, and *bcl2*) showing divergent expression during embryogenesis (**G**) but co-expression during regeneration (**H**). **C, F** GO-term enrichment was assessed using a one-sided Fisher's exact test (*P* < 0.02). **D, E, G, H** Data was extracted from the NvERTx database[39].

mesenterial Tips (AG/AMT: *runt, moxD, axin-like, twist, wls;* Fig. 3Ch–l), amputation site gastrodermis + mesenteries (AG/M: *sprouty, musk-like, porcupine-like, smad4-like;* Fig. 3Cm–p), mesenteries (M, *hd50, mae-like, bicaudal-like, k50-5, pdvegfr-like, fox1, foxD1, mox1, phtf1-like;* Fig. 3Cq–y) and the physa (P, *fgfA2;* Fig. 3Cz). Taken together with their temporal expression profiles reflected by their expression clusters (Fig. 3B; Supplementary Data 7, 8, and 9), most of these genes form synexpression groups[53], potentially involved in the same biological

process and underlying gene regulatory networks within their respective expression domains.

## Injury-induced regeneration activates apoptosis, pERK, and cWnt

Having observed a strong enrichment for apoptosis related GO-terms in the list of 124 regeneration-specific genes and in the regeneration specific module R6, we investigated the dynamics of apoptosis during

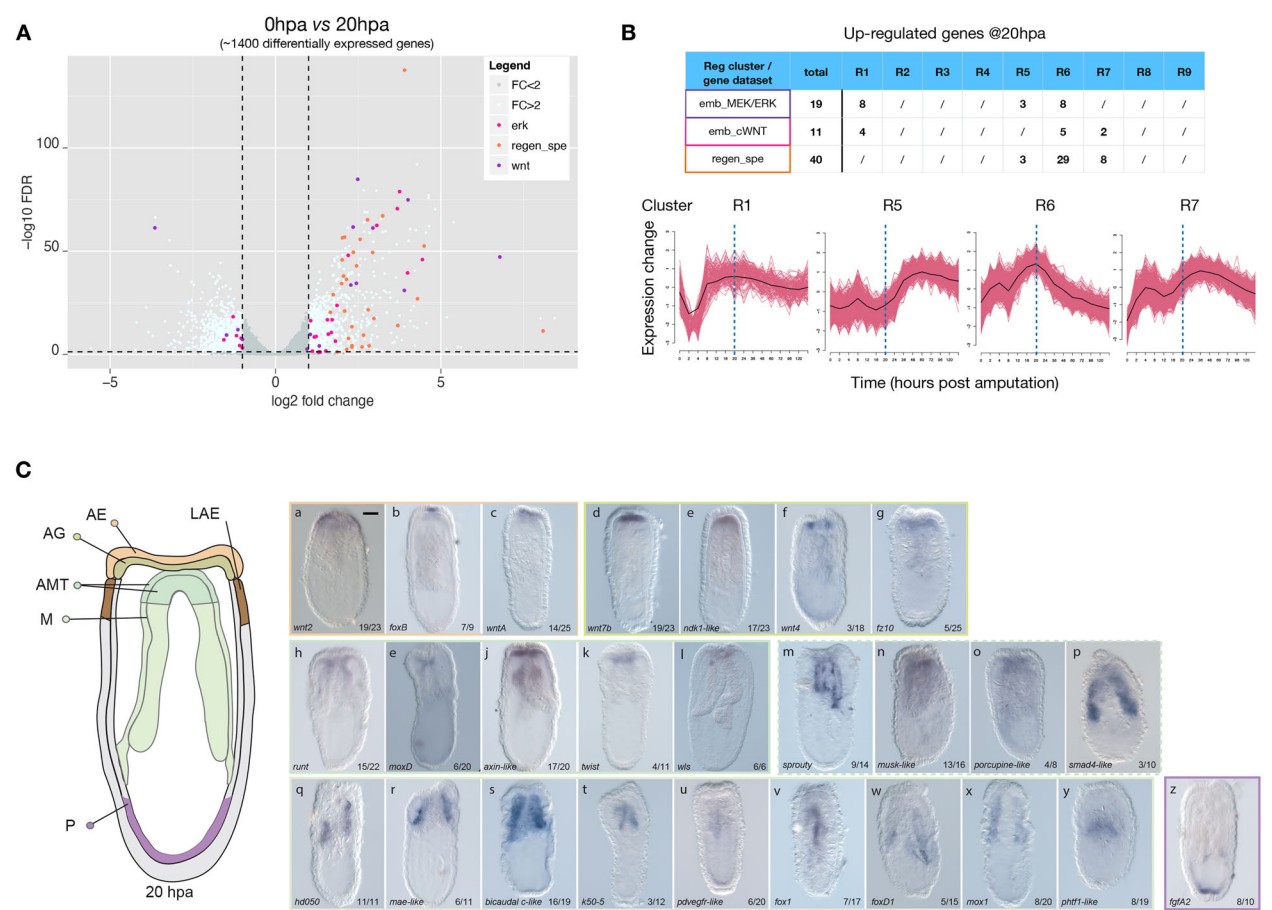

**LAE** - Lateral amputation site ectodermis, **AE** - Amputation site ectodermis, **AG** - Amputation site gastrodermis, **AMT** - Amputation site mesenterial tips, **M** - Mesenteries, **P** - Physa

**Fig. 3 | Identification and characterization of genes upregulated at 20 hpa.**
**A** Volcano plot highlighting the ~1400 differentially expressed genes (DEGs) (log2(FC)| > 2 & FDR < 0.05) at 20 hpa (from ref. 39). Red, orange and purple dots indicate genes that belong to the embryonic MEK/ERK (emb_MEK/ERK) or cWnt (emb_cWnt) downstream targets, as well as the regeneration specific genes (regen_spe), respectively. **B** Overview of the up-regulated genes at 20 hpa that belong to either of the three datasets (emb_MEK/ERK, emb_cWnt or regen_spe) highlighting the regeneration expression clusters they are assigned to. The same overview for the down-regulated genes can be found in Fig. S3. **C** Schematic

representation of the various expression domains (LAE lateral amputation site ectoderm, AE amputation site ectoderm, AG amputation site gastrodermis, AMT amputation site mesenterial tips, M Mesenteries and P physa) characterizing the regenerating polyp as revealed by whole-mount in situ hybridization at 20 hpa. (Ca-z) Expression patterns of indicated genes (bottom left) at 20 hpa. The colored rectangles surrounding the brightfield images, correspond to the expression pattern color code indicated in the schematic representation. The number in each panel (bottom right) indicates the total number of polyps with the represented pattern. Scale bar (first picture, upper right corner): 20 μm.

the regenerative process. Several genes relating to apoptosis, including the regeneration-specific genes *bax* (NvERTx.4.118493), *caspase-3* (NvERTx.4.152711), *bcl2* (NvERTx.4.81296), and an additional *bcl2* (which we term *bcl2B*; NvERTx.4.84383), belong to the regeneration specific cluster R6 and are activated shortly after amputation (Fig. 4A; Supplementary Fig. S4A). Whole mount in situ hybridization for *bax* 24 hpa confirms the increased dynamics and indicates that like other genes from the expression cluster R6 it is activated at the amputation site as early as 2 hpa and progressively increases its expression in the mesenteries in later stages (Fig. 4C; Supplementary Fig. S4B).

We next performed a time series of TUNEL staining to examine the dynamics of apoptosis during embryogenesis and regeneration. While only few apoptotic cells can be observed during embryonic development[54], during regeneration we observed a burst of apoptotic activity after amputation at the cut site as early as 1.5 hpa which perdured through 12 hpa (i.e., 1st apoptotic wave; Fig. 4D; Supplementary Figs. S4 and S5). At 24 hpa apoptotic activity is now randomly detected along the body wall epithelia of the entire polyp, potentially involved in the systemic response (i.e., 2nd apoptotic wave; Fig. 4D; Supplementary Fig. S4). This dynamic TUNEL profile is reminiscent of the *bax* expression pattern.

In addition to apoptosis, we assessed the activation of MEK/ERK and cWnt signaling pathways during the onset of regeneration. This choice has been made to compare the regeneration GRN with the GRN underlying embryonic development that is driven by MEK/ERK/Erg and cWnt/Tcf signaling[26,51,52]. *erg* (NvERTx.4.84016) and *tcf* (NvERTx.4.100051) are the respective MEK/ERK and cWnt effectors involved in *N. vectensis* embryonic germ layer formation[26,51,52]. Analyzing their expression pattern during regeneration, revealed that both genes are up-regulated after wound-healing (Fig. 4E, I) and localized at the *TC* between the oral part of the mesenteries and the body wall epithelia at the amputation site (Fig. 4G, K; Supplementary Figs. S2 and S6).

Using a monoclonal antibody directed against the phosphorylated/activated form of ERK (pERK) during the time course of regeneration (Fig. 4H; Supplementary Fig. S6), we detected localized pERK staining as early as 1 hpa in the body wall epithelia at the amputation site as well as in the oral tips of the mesenteries (Fig. 4H). After wound-healing (6 hpa, Fig. S6), this staining remains localized as regeneration progresses until 48 hpa (Fig. 4H). A localized signal is detected in the reforming pharynx (72, 96 hpa), after which the signal begins to be re-detected ubiquitously throughout the body (120, 144 hpa),

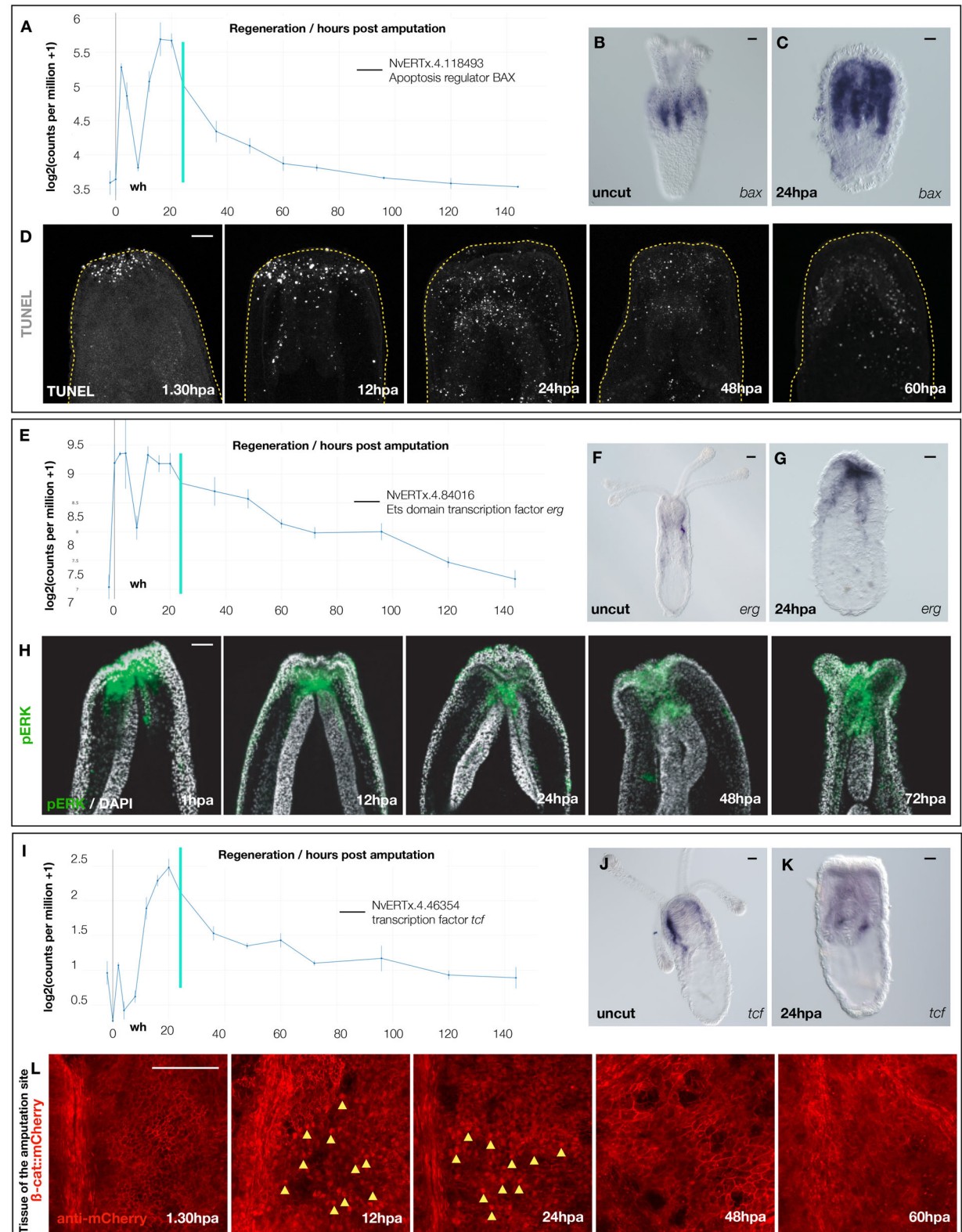

**Fig. 4 | Apoptosis, pERK and cWnt are activated in response to injury.** Temporal expression pattern for *bax* (**A**), *erg* (**E**) and *tcf* (**I**) following sub-pharyneal amputation and during regeneration (data extracted from NvERTx.ircan.org[39]. Spatial expression pattern for *bax* (**B**, **C**), *erg* (**F**, **G**) and *tcf* (**J**, **K**) in uncut cut polys at 24 hpa. **D** TUNEL staining (white dots) during regeneration (1.5–60 hpa). The yellow dashed line indicates the outline of the polyp with the oral part/amputation site to the top. **H** Immunohistochemistry using an anti-pERK antibody revealing the activated/ phosphorylated form of ERK (green) in uncut and regenerating polyps from 1 to 72 hpa (counterstaining: DAPI). **L** Immunohistochemistry using an anti-mCherry antibody on regeneration Nvß-catenin::mCherry polyps. Confocal stacks focussing on the amputation site. Nuclear staining (yellow arrows) indicates the nuclearized and activated form of ß-catenin. Scale bar (upper right corner): 20 μm. **A**, **E**, **I** Data are extracted from the NvERTx database (from ref. 39).

reminiscent of the staining detected in uncut controls (Fig. 4H; Supplementary Fig. S6).

To determine localization of cWnt activation, we developed a transgenic knock-in line in which we fused the fluorescent reporter mCherry in frame with the C-terminal end of the endogenous copy of ß-catenin (Fig. S7). Positive F1, *Nvß-catenin::mCherry*, were outcrossed and offspring used to assess early cleavage stages enabling us to observe nuclearization of ß-catenin (Movie S1) confirming previous observations[25–27,30]. We then assessed ß-catenin nuclearization in *Nvß-catenin::mCherry* polyps during the process of regeneration using an anti-mCherry antibody (Fig. 4L). While no nuclear staining was observed at 1h30 hpa, nuclear staining became detectable at 12 hpa and persisted at least up to 24 hpa at the amputation site. Intriguingly, we did not detect any nuclear staining at 48 hpa and 60 hpa.

Taken together these spatio-temporal expression and activation patterns suggest that MEK/ERK signaling, and apoptosis are activated shortly after amputation and during the process of regeneration, while cWnt signaling is activated only during regeneration after wound-healing is completed.

## Regeneration in *N. vectensis* requires apoptosis, MEK/ERK, and cWnt

To test whether apoptosis, MEK/ERK and/or cWnt are required for regeneration, we treated amputated juveniles during the time course of this process with pharmacological inhibitors reported and confirmed to be efficient in *N. vectensis*; the pan-caspase inhibitor Z-VAD[55] to block apoptosis, the MEK inhibitor U0126[51,52,56] and iCRT14 to block cWnt signaling[57,58] (Figs. S8 and S9).

Previous studies have shown that perturbation of MEK/ERK and cWnt block embryonic development and patterning in *N. vectensis*[26,27,51,52,56]. We therefore tested whether apoptosis is a regeneration-specific process or if it was also required for embryonic development. Interestingly, *N. vectensis* embryos treated continuously with Z-VAD after fertilization developed normally into functional polyps, showing no developmental defect and metamorphosed on time as in controls (Fig. 5A). While Z-VAD treated animals were slightly smaller than controls (Fig. S10), the absence of visible phenotypes during embryonic development and metamorphosis support the idea that apoptosis in *N. vectensis* is a regeneration-specific process when compared to embryonic development.

We then treated regenerating polyps continuously immediately after amputation with either Z-VAD, U0126 or iCRT14 up to 144 hpa. Although all treatments impaired the regeneration program, there were visible differences (Fig. 5B). Z-VAD treatments prevented the emergence of TUNEL positive (TUNEL+) cells as soon as 1.5 hpa (Fig. S8A) and regeneration was blocked in an early regenerative stage (i.e., modified stage 1), lacking the physical interaction between the fused oral tips of the *MES* and the *BWE* at the wound site (Fig. 5Bb). U0126 treatments effectively blocked pERK activation as early as 1 hpa (Fig. S8B) and animals were blocked in an even earlier regeneration stage (stage 0.5), i.e., neither the fusion of the *MES* tips nor the *TC* between the *MES* and the *BWE* of the wound site were observed (Fig. 5Bc). iCRT14 treated polyps did not display ß-catenin nuclearization at 24 hpa (Fig. S8C) and in line with a later activation of the cWnt pathway described above, treated polyps regenerated up to stage 2 but never formed a pharynx or tentacles (Fig. 5Bd).

Inhibition of MEK/ERK signaling (using U0126) or apoptosis (using Z-VAD) from 0 to 144 hpa both block the regeneration process early on. To gain further insight into the roles of these two pathways we performed waves of inhibitory treatments during various time windows and assessed the resulting phenotypes (Fig. S11). When U0126 treatments were started at 24 or 48 hpa, and lasted up to 144 hpa, regeneration was blocked at stage 2 in most cases (100% $n = 75/75$ and 75% $n = 50/75$, respectively) (Fig. S11A). When U0126 was added at 72 hpa or later time points up to 144 hpa, regeneration was completed

successfully or slightly delayed in comparison to DMSO treated controls. These results indicate that MEK/ERK signaling is crucial as early as stage 0.5 for initiating a regenerative response.

To dissect the temporal contribution of apoptosis during regeneration, we selectively inhibited each of the three apoptotic waves using time-specific Z-VAD treatments and evaluated regeneration outcomes at 144 hpa (Fig. S11B) following the staging system from ref. 36. Independent inhibition of the first apoptotic wave (0–6 hpa), the second wave (12–48 hpa), or the third wave (48–144 hpa) resulted in a substantial proportion of polyps successfully completing regeneration, i.e., 73% ($n_{total} = 78$), 53% ($n_{total} = 70$), and 53% ($n_{total} = 68$), respectively. While the majority of polyps reached stage 4, some showed defects in tentacle formation—especially when apoptosis was blocked during the second (32%) or third (72%) waves. These findings suggest that while apoptosis during each wave contributes to overall regenerative success, apoptosis during the later stages may play a more critical role in the proper morphogenesis of specific structures, such as tentacles.

## MEK/ERK signaling is required for injury-induced apoptosis and cWnt

To gain further insight into the different regeneration phenotypes caused by the treatments, we assessed their effects on earlier hallmarks, such as wound-healing (Fig. 5C). Wound-healing is terminated in about 90% ($n_{total} = 48$) of control polyps by 6 hpa[36] (Fig. 5C). While none of the treatments blocked wound-healing, it is nonetheless noticeable that this process is significantly delayed in U0126 treatments (wound is healed at 8hpa in 90%, $n_{total} = 36$), but significantly accelerated in Z-VAD and iCRT14 treatments, i.e., wound is healed at 4hpa in 100% ($n_{total} = 27$) and 90% ($n_{total} = 28$), respectively (Fig. 5C). To determine whether this delayed wound-healing is responsible for the arrest in the regeneration process, we compared the effects of U0126 treatments starting at 0hpa or 8 hpa (i.e., after wound-healing is completed[36]) on their capacity to reach at least stage 1 at 24 hpa. While most control animals (12 out of 15 cases) reached stage 1 at 24 hpa, U0126-treated animals were consistently blocked at stage 0.5 (start 0 hpa: 14 out 16 cases; start 8 hpa: 12 out of 13 cases) (Fig. S12), indicating that regeneration is blocked independently of the delayed wound-healing process.

*TC* between the *MES* and the body wall epithelia *BWE* at the amputation site is a hallmark of regeneration in *N. vectensis* and observable at 24 hpa and 48 hpa[36]. In a similar timing, the initiation (24 hpa) and burst (48 hpa) of amputation-induced cell proliferation occurs at the amputation site[34,36]. We thus performed inhibition treatments between 0–24 hpa, 0–48 hpa or 24–48 hpa, and assessed their impact on the initiation or maintenance of the TC (Fig. 5D, Supplementary Data 10) and cell proliferation (Fig. 5E). In animals treated with Z-VAD we observed that the initiation of the TC occurred in a similar manner than in controls at 24 hpa (Fig. 5Dd). However, its maintenance was impaired at 48 hpa (Fig. 5De, f). More drastically, inhibition of MEK/ERK prevented not only the initiation of TC at 24 hpa (Fig. 5Dg) but also its maintenance at 48 hpa (Fig. 5Dh, i). In contrast, when cWnt signaling was inhibited TC was simply delayed at 24 hpa and maintained at 48 hpa (Fig. 5Dj–l). Further quantification of the number of cells in S-Phase (EdU+ cells) in these same conditions, revealed a significant down-regulation of cell proliferation at the end of all treatment windows (Fig. 5D, E and Supplementary Data 11).

Taken together, these data indicate that injury-induced initiation and maintenance of cell proliferation depends on a constructive function of apoptosis as well as MEK/ERK and cWnt signaling, while the impact of these pathways on the initiation and maintenance of the TC is pathway dependent. Based on this global phenotype assessment, MEK/ERK signaling emerged as to have the earliest impact with the broadest range of phenotypes linked to the initiation of the regeneration program in *N. vectensis* (Fig. 5F). These observations further

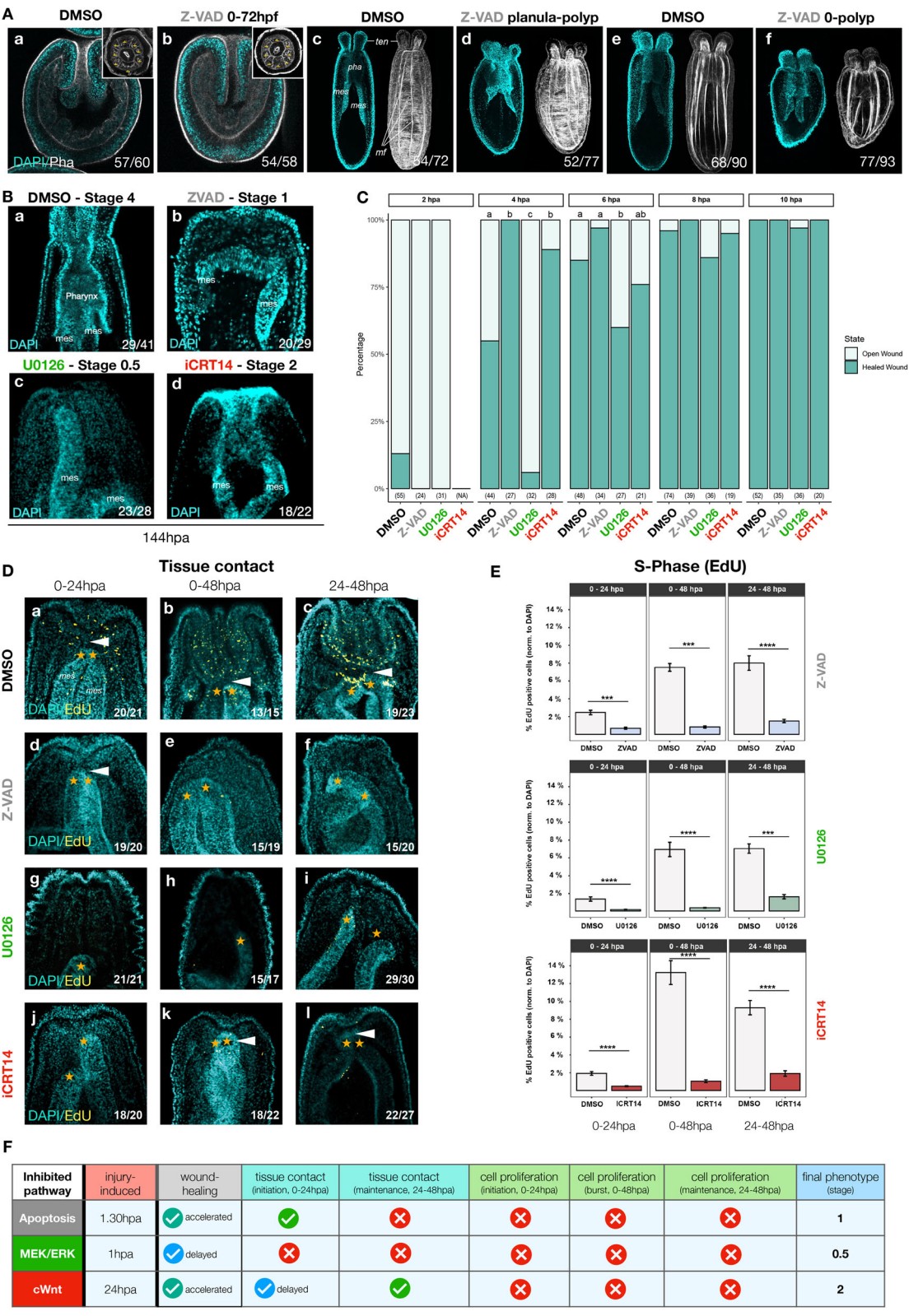

indicate that the premature (stage 0.5) arrest of regeneration upon inhibition of MEK/ERK occurs through the inhibition of morphogenetic movements during the proliferation-independent phase of regeneration[34,36].

We next sought to understand the potential relationship between these three signaling pathways at 20 hpa (Fig. 6). While Z-VAD treatments had no visible effects on TUNEL staining or the activation of

pERK, nuclearization of ß-catenin appeared slightly increased compared to control conditions (Fig. 6A). Following U0126 treatments, all three pathways seemed impaired which was reflected by the absence of TUNEL+ cells, pERK staining and nuclearization of ß-catenin (Fig. 6A). Finally, iCRT14 prevented nuclearization of ß-catenin, did not affect pERK staining but increased the amount of TUNEL+ cells (Fig. 6A). Taken together, these results suggest that MEK/ERK signaling

**Fig. 5 | Apoptosis, MEK/ERK and cWnt are required for regeneration.**
**A** Confocal stacks of developing late gastrula (a, b) and post-metamorphic polyps (c–f). Treatments with DMSO (**A**a, c, e) or the pan-caspase inhibitor Z-VAD exposed from planula do not affect embryonic development or metamorphosis. Gastrulae and polyps are stained with Phalloidin (f-actin in white) and DAPI (nuclei in turquoise). *pha*: pharynx; *mes*: mesenteries; *mf*: muscle fibers; *ten*: tenatcles.
**B** Confocal stacks of regenerating polyps at 144 hpa. Treatments with DMSO (**A**a) or the pan-caspase inhibitor Z-VAD (**A**b), the MEK inhibitor U0126 (**A**c) or the cWnt inhibitor iCRT14 (**A**d) that block regeneration at stage 1, 0.5 or 2, respectively. Polyps are stained with DAPI (nuclei in turquoise). mes: mesenteries. Numbers in the right bottom corner indicate the case number of the shown phenotype.
**C** Diagram of results obtained from the compression/wound-healing assays[36] comparing DMSO, Z-VAD, U0126, or iCRT14 treated polyps during the first 10 h following sub-pharyngeal amputation. Light and dark green bars indicate an open or healed wound, respectively. Numbers in parenthesis indicate the number of assessed animals. Different letters above the bars indicate statistically significant

differences among the different treatment ($p < 0.05$) according to two-sided pairwise Fisher's exact tests with Benjamini–Hochberg adjustment. For each condition, biological triplicates were pulled. **D** Confocal stacks of regenerating polyps counterstained with DAPI (cyan) and EdU (yellow) focused on the TC (white arrowhead) at 24 hpa or 48 hpa following treatment with DMSO (**C**a–c), Z-VAD (**C**d–f), U0126 (**C**g–i) or iCRT14 (**C**j–l) for 0–24 hpa, 24–48 hpa or 0–48 hpa. Orange stars indicate the oral part of the mesenteries. Numbers in the right bottom corner indicate the case number of the shown phenotype. For each condition biological triplicates were pulled. **E** Graphical representation of the EdU counts (yellow staining in (**D**)) extracted from the confocal stacks of the treatments performed in (**D**). For each condition 10 animals were selected randomly for EdU counting. Data are presented as mean values ± SEM. Statistical significance was determined using pairwise two-sided Wilcoxon–Mann–Whitney tests (non-parametric). *** indicates *p* value < 0.0001 **** indicates *p* value < 0.00001. **F** Summary of the observed phenotypes. Statistical analyses are provided in Sup Table 26. Source data are provided as a Source Data file.

is up-stream of apoptosis and cWnt signaling and that apoptosis and cWnt signaling are required for mutual repression during the onset of regeneration (Fig. 6B).

## MEK/ERK, cWnt, and apoptosis have largely distinct downstream targets

To gain further insight into the molecular mechanisms activated by MEK/ERK, cWnt or apoptosis at the onset of regeneration, we performed bulk RNAseq for each of these treatments at 20 hpa (Fig. 7A and Supplementary Data 12) in adults that display the same phenotypes as juveniles (Fig. S13 and Supplementary Data 13). We assessed significantly differentially expressed genes ($\log_2FC > 1$ or $>-1$ and an adjusted $p < 0.05$) in any treatment compared to control/DMSO-treated regenerating animals. We identified 342 (193 up/149 down), 578 (304 up/274 down) and 472 (220 up/252 down) genes that were differentially expressed in Z-VAD, U0126 and iCRT14-treated animals, respectively (Fig. 7B; Supplementary Data 14, 15, and 16).

Strikingly, GO-term enrichment analysis of these differentially expressed genes (Fig. 7C and Supplementary Data 17) largely confirmed the observations made following the experiments to address the functional relationships between these pathways. In line with the observation that MEK/ERK signaling controls cWnt signaling and apoptosis, GO-terms enriched in the dataset of genes down-regulated by U0126 at 20 hpa contained Wnt signaling and apoptosis. Furthermore, and in line with the observation that cWnt and apoptosis are required for mutual repression during the onset of regeneration, GO-terms enriched in the dataset of genes up-regulated by Z-VAD or iCRT14, contain Wnt signaling and neuron death, respectively (Fig. 7C and Supplementary Data 17). Assessing the temporal expression patterns of the genes down- or up-regulated following the various treatments (Fig. S14) further supports the central role of MEK/ERK signaling during the onset of regeneration. In fact, while most of the apoptosis and cWnt downstream targets are mainly up-regulated either shortly after amputation or late during the regeneration process, a large part of MEK/ERK downstream targets are also activated at the onset of regeneration, i.e., 12–24 hpa (Fig. S14).

We then compared the lists of differentially expressed genes and observed that MEK/ERK, cWnt and apoptosis during regeneration in *N. vectensis* appear to have largely distinct downstream targets (Fig. 7D; Supplementary Data 18, and 19). In fact, only a small number of down-regulated genes were shared between the different treatment conditions. Specifically, five genes were commonly downregulated in all three treatments. When comparing pairs of treatments, eight genes were shared between U0126 and Z-VAD, 20 genes between U0126 and iCRT14, and 28 genes between Z-VAD and iCRT14 (Fig. 7D and Supplementary Data 18). Similarly, only a limited number of up-regulated genes were shared across treatments. Specifically, 10 genes were commonly upregulated in all three treatments. Pairwise comparisons

revealed 14 shared genes between U0126 and Z-VAD, 39 genes between U0126 and iCRT14, and 45 genes between Z-VAD and iCRT14. (Fig. 7D and Supplementary Data 19).

The absence of strong overlap between the downstream targets of all three pathways might appear surprising because UO126 treatments affect both apoptosis and cWnt activation (Fig. 6Ac, k). The absence of overlap between U0126 and iCRT14 can be explained by the delayed *TC* in iCRT14-treated animals (Fig. 5Dj), the presence of compensatory regulatory mechanisms that drive cWnt target genes during regeneration in *N. vectensis* independently of MEK/ERK signaling, or spatially restricted actions of MEK/ERK affecting injury-induced cWNT. The absence of overlap between U0126 and Z-VAD can be explained by the caspase-independent activation of apoptosis at 20 hpa (Fig. 6Ab). Taken together, these data suggest that although MEK/ERK is required for apoptosis and cWnt signaling at 20 hpa, these three pathways seem to act at different moments in a complementary manner during the regeneration process in *N. vectensis*.

## MEK/ERK regulates "embryonic" and "regeneration-specific" genes

We next sought to understand to what extent either of these three pathways are controlling injury-induced expression of the 15 embryonic cWnt (emb_cWnt) and 25 MEK/ERK (emb_ERK) downstream targets as well as the 84 regeneration-specific (regen_spe) genes up- or down-regulated at 20 hpa (Fig. 3). Systematically comparing these pools of genes (emb_cWnt, emb_ERK, regen_spe) with the up- or down-regulated genes following inhibition of MEK/ERK (Fig. 7; Supplementary Data 20 and 21), apoptosis (Fig. 7; Supplementary Data 22 and 23), or cWnt (Fig. 7; Supplementary Data 24 and 25) during regeneration, we observed that MEK/ERK signaling during regeneration has the most widely extended control on gene expression for all three pools (Fig. 8 and Supplementary Fig. S15). This is mostly visible in the down-regulated genes (Fig. 8A). In fact, no genes of the "embryonic" pools are affected by neither Z-VAD nor iCRT14, and only three genes from the regeneration specific genes are downregulated by iCRT14. However, six, seven, and eight genes belonging to the embryonic ERK, embryonic cWnt or regeneration-specific gene pools, respectively, are downregulated by U0126 treatments at 20 hpa (Fig. 8A and Table 1).

To further confirm that these above-identified genes are downstream targets of MEK/ERK during regeneration, we treated animals continuously after sub-pharyngeal amputation or starting at 8hpa (after wound-healing) with U0126 and analyzed the expression levels of the three gene pools by RT-qPCR at 20 hpa (Fig. 8B; Supplementary Fig. S16). 19 out of the 25 identified embryonic MEK/ERK downstream targets are downregulated by U0126 treatments during regeneration (Fig. 8B). Similarly, 11 out of the 15 identified embryonic cWnt downstream targets are also downregulated following U0126 treatment (Fig. 8B). Finally, for the "regeneration-specific" pool of genes we only

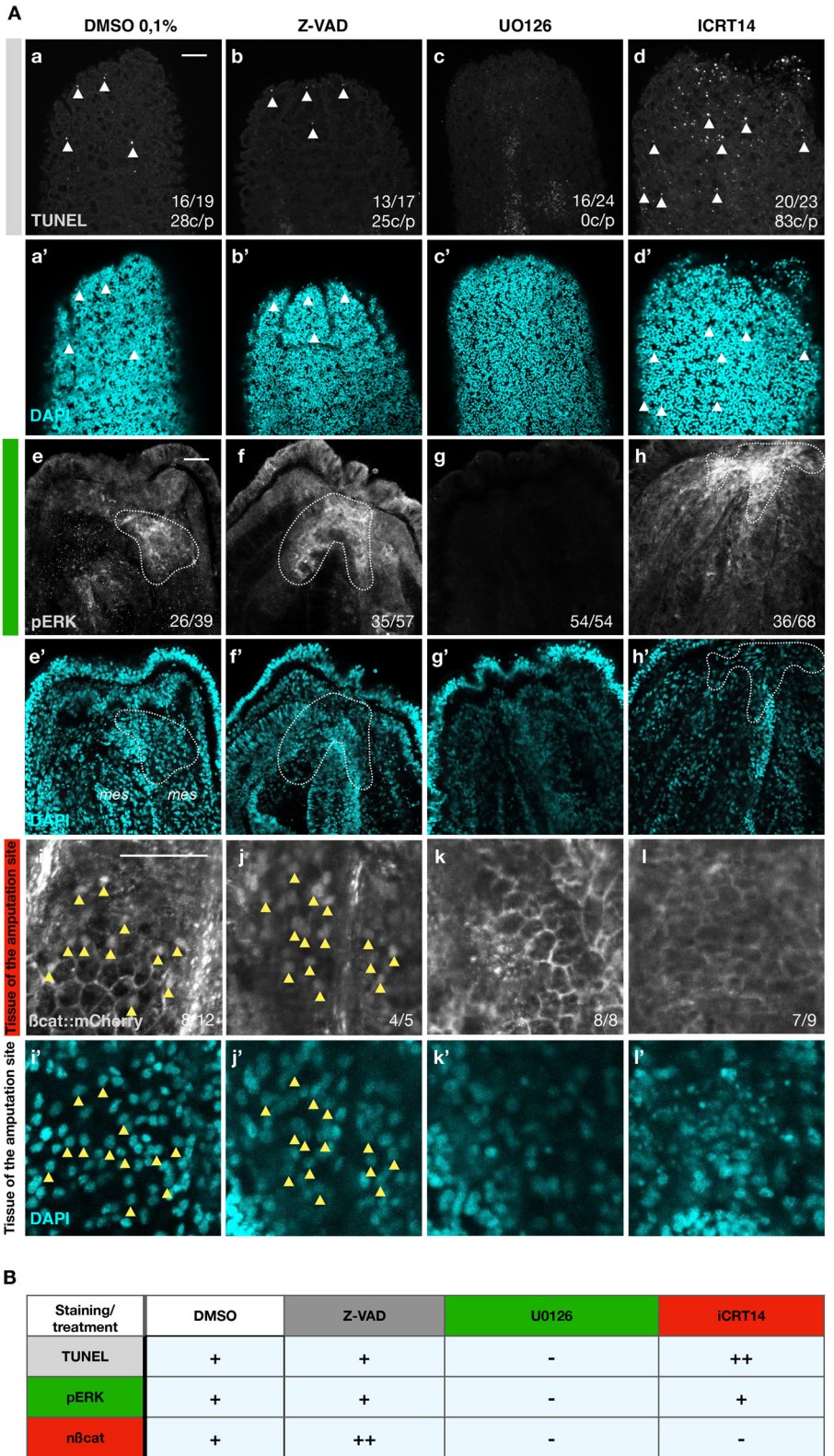

**Fig. 6 | MEK/ERK is required for apoptosis and cWnt signaling. A** Confocal stacks of regenerating polyps following DMSO (a, e, i), Z-VAD (b, f, j), U0126 (c, g, k) or iCRT14 (d, h, l) treatments. (a'–l') DAPI (nuclear) counterstaining of the corresponding stacks stained with TUNEL (a–d), pERK (e–h) or mCherry (i–l). Numbers in the bottom right corner indicate the case number of the shown phenotype. White arrowheads indicated TUNEL+ cells, dashed lines indicate pERK positive areas and yellow arrowheads indicated nuclearized ß-catenin. Scale bar (upper right corner): 20 µm. **B** Summary of the observed phenotypes. Colored bars and boxes indicate pathway-specific treatments and read-outs.

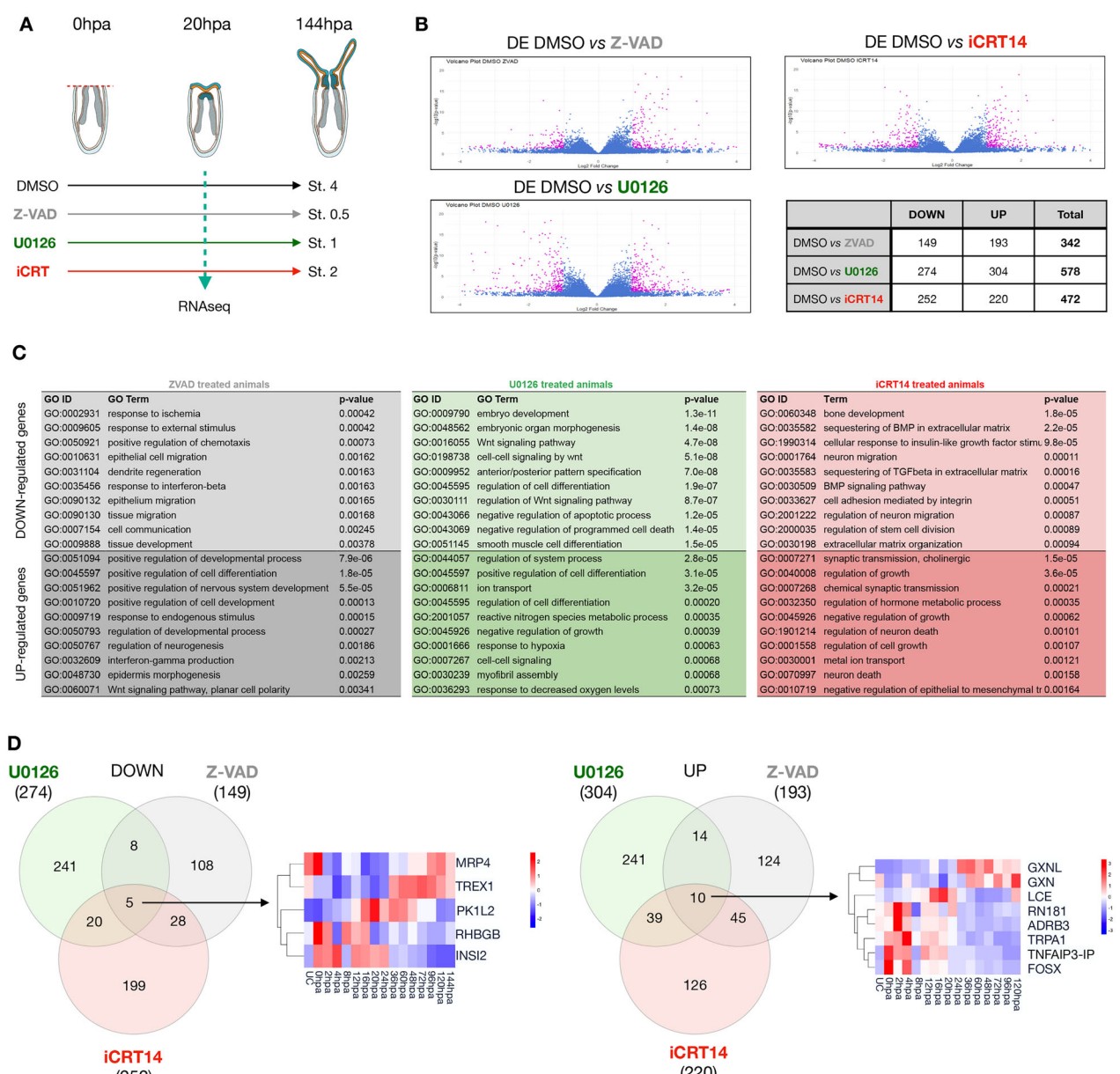

**Fig. 7 | Apoptosis, MEK/ERK and cWnt have largely distinct downstream targets during regeneration. A** Overview of the experimental design. **B** Volcano plots and summary table highlighting the DEGs (|log2(FC)| >2 & $p < 0.05$) for each treatment compared to DMSO at 20 hpa. **C** Selected Gene Ontology (GO) terms are shown for genes that were up- or down-regulated by Z-VAD, U0126 or iCRT14 at 20 hpa. GO-term enrichment was assessed using a one-sided Fisher's exact test ($P < 0.02$). Extended GO-term lists for each treatment are available in Fig. S14, Supplementary Data 17. **D** Venn diagrams illustrating the comparison of the down- and up-regulated DEGs for Z-VAD (gray), U0126 (green) and iCRT14 (red). Heatmaps represent the temporal expression pattern during regeneration of the pool of genes down- or up-regulated by all three treatments. Source data are provided as a Source Data file.

assessed 26 out of the 40 identified genes of which 19 were down-regulated when MEK/ERK signaling was blocked by U0126 (Fig. 8B). This analysis not only largely confirms but also extends the list of the above described MEK/ERK downstream targets identified by RNAseq (Fig. 8A and Table 1). Furthermore, we also assessed the effects of MEK/ERK inhibition on *erg* and *tcf* (insets Fig. 8B), two transcription factors that are effectors of MEK/ERK and cWnt, respectively, during embryonic development in *N. vectensis*[26,51]. Interestingly, we observed a reduced expression for *erg*, suggesting a negative feedback loop, and a strong down-regulation of *tcf* under those conditions.

In situ hybridization of uncut, 20hpa DMSO treated and U0126 treated juveniles for a subset of these genes (*sprouty, runt, mae-like,*

*erg; twist, foxB, axin-like, tcf; carm1, cad, egln1,* and *bax*), confirmed the RT-qPCR data (Fig. 8C). This analysis also revealed that at the onset of regeneration, MEK/ERK signaling controls expression of genes in several domains of the amputation site, including the AE, AG, AMT and M. Combining the obtained data from RNAseq, RT-QPCR (i.e., temporal expression), and in situ hybridization (i.e., spatial expression) as well as functional studies together, these results enabled us to propose a MEK/ERK-dependent GRN blueprint underlying whole-body regeneration in *N. vectensis* (Fig. 8D). This GRN blueprint indicates the potential direct or indirect downstream targets of MEK/ERK signaling at the onset of regeneration, that (i) belong either to the group of embryonic cWnt or embryonic MEK/ERK targets (including their

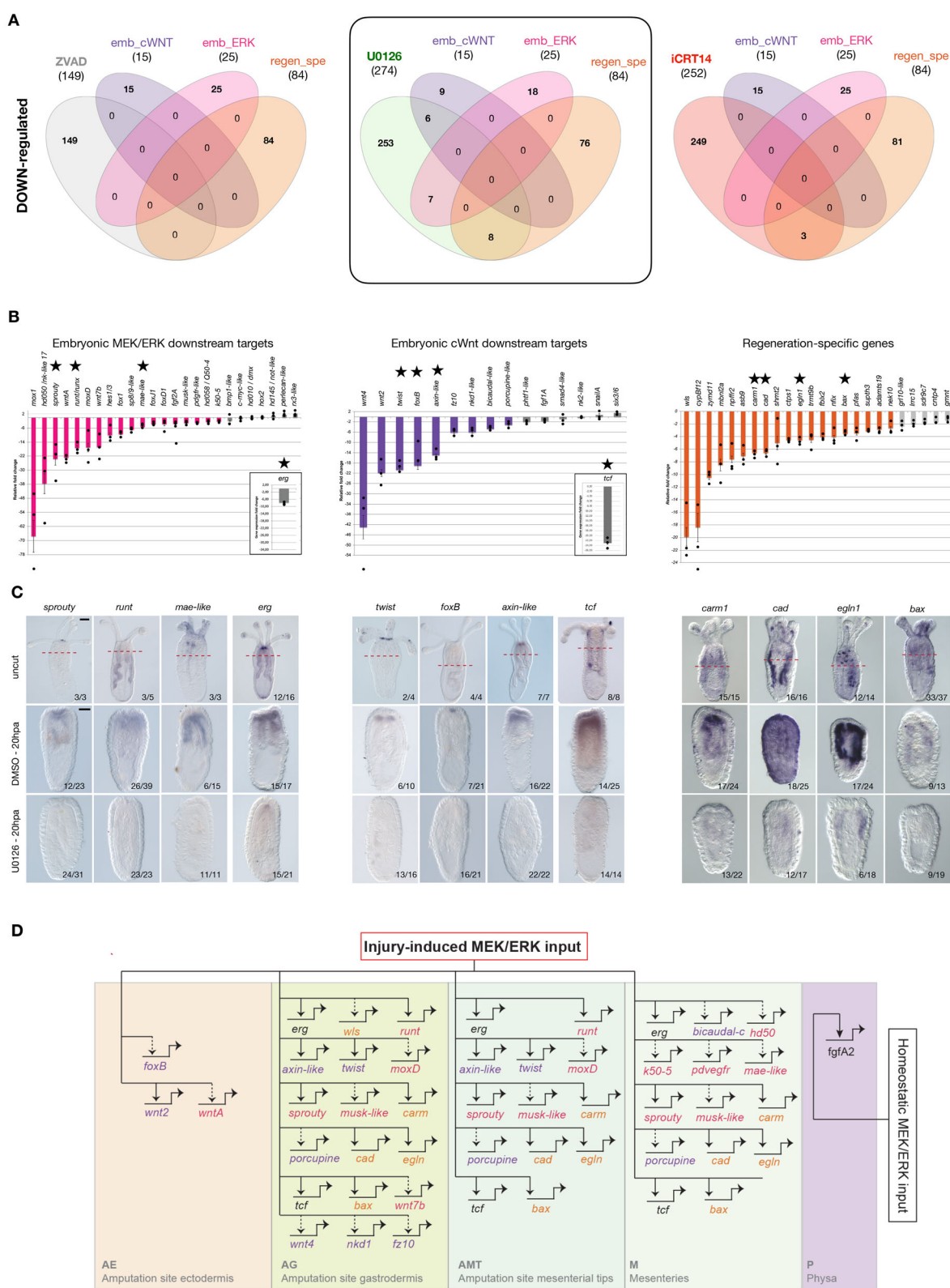

respective potential effector genes *tcf* and *erg*) and (ii) are interconnected with "regeneration-specific" genes.

## Discussion

In this study we used whole genome transcriptomic profiling to identify shared embryonic and regeneration-specific gene signatures. Comparing dynamically expressed genes in both processes, this approach revealed that regeneration appears transcriptionally modest compared to embryogenesis (Fig. 1). The differences in transcriptional dynamics might be considered in contrast to recent studies carried out in sponges and arthropods that showed that a similar number of genes is dynamically expressed during post-larval development and

**Fig. 8 | MEK/ERK controls embryonic cWnt and MEK/ERK downstream targets as well as regeneration-specific genes during regeneration. A** Venn diagrams illustrating the comparison of the down-regulated DEGs from either Z-VAD, U0126 or iCRT14 with the emb_cWnt, emb_ERK or regeneration-specific (regen_spe) datasets. The square indicates that U0126 treatments affect expression of genes in all three datasets. Venn diagrams illustrating the corresponding down-regulated DEGs can be found in Fig. S15. **B** RT-qPCR analysis at 20 hpa comparing DMSO and U0126 treated polyps with genes identified as being transcriptionally upregulated at 20 hpa and corresponding either to "embryonic MEK/ERK downstream targets", embryonic cWnt downstream targets", or "regeneration-specific" genes (including those identified to be potential MEK/ERK downstream targets). Data are presented as mean values ± SEM and dots represent FC values of replicates. Source data is provided as a Source Data file. Colored bars in each panel indicate genes that were at least 2-fold downregulated by U0126 treatments, while gray bars indicate genes unaffected by the inhibition of MEK/ERK signaling during regeneration. The inserts

indicate the RT-qPCR for *erg* and *tcf*, both significantly down and the potential effectors of the MEK/ERK and cWnt pathways, respectively. **C** Genes with a black star were further analyzed by whole mount in situ hybridization following U0126 treatments. The 1st row corresponds to uncut DMSO treated controls with the amputation site indicated in dashed orange lines. The 2nd row corresponds to DMSO treated controls at 20 hpa and the 3rd row are U0126-treated regenerating polyps fixed at the same time (20 hpa). Numbers indicate the ratio of polyps with the shown expression pattern/total polyps. **D** Blueprint of the MEK/ERK-dependent GRN underlying the onset of regeneration (20 hpa). Colors of the boxes represent the expression patterns as shown in Fig. 3C. Genes are placed within this GRN according to their expression patterns in either one or several boxes. Solid lines indicate functional evidence that MEK/ERK signaling controls expression of the downstream target obtained by RT-qPCR and verified by in situ hybridization. No assumption on whether these interactions are direct or indirect is made. Scale bar (first pictures of **C**, upper right corner): 20 μm.

## Table 1 | Genes belonging either to the embryonic cWnt (emb_cWnt), embryonic MEK/ERK (emb_ERK) downstream targets or to the regeneration specific (regen_spe) datasets and downregulated by U0126 at 20 hpa

| Overlap | NvERTx ID | UniProt | Cluster |
|---|---|---|---|
| U0126/emb_cWnt | NvERTx.4.116634 | AXIN1 | R1 |
| U0126/emb_cWnt | NvERTx.4.48248 | WNT4 | R1 |
| U0126/emb_cWnt | NvERTx.4.228252 | FOXB | R7 |
| U0126/emb_cWnt | NvERTx.4.114443 | WNT2 | R6 |
| U0126/emb_cWnt | NvERTx.4.219905 | TWIST | R1 |
| U0126/emb_cWnt | NvERTx.4.228082 | FZ10 | R1 |
| U0126/emb_ERK | NvERTx.4.122455 | OTXB | R1 |
| U0126/emb_ERK | NvERTx.4.6665 | FOX1 | R1 |
| U0126/emb_ERK | NvERTx.4.77035 | MOX1 | R1 |
| U0126/emb_ERK | NvERTx.4.77043 | MOX2 | R5 |
| U0126/emb_ERK | NvERTx.4.132141 | WNTA | R1 |
| U0126/emb_ERK | NvERTx.4.82946 | MAE | R6 |
| U0126/emb_ERK | NvERTx.4.64399 | SPRY | R6 |
| U0126/regen_spe | NvERTx.4.220399 | TLX3 | R5 |
| U0126/regen_spe | NvERTx.4.165999 | CARM1 | R6 |
| U0126/regen_spe | NvERTx.4.161569 | ADAMTSL1 | R6 |
| U0126/regen_spe | NvERTx.4.220920 | ADAMTS19 | R6 |
| U0126/regen_spe | NvERTx.4.3403 | CTPS1 | R6 |
| U0126/regen_spe | NvERTx.4.52541 | No description | R9 |
| U0126/regen_spe | NvERTx.4.154731 | No description | R4 |
| U0126/regen_spe | NvERTx.4.213276 | SLC20A1 | R5 |

regeneration (sponge[19]) as well as post larval limb development and regeneration (arthropod[22]). However, it is important to point out that our study also included early developmental stages and not only those following larval development, therefore covering a larger set of developmental processes. This may also be the reason why in *N. vectensis* we observed a drastic difference of transcriptional dynamics between embryonic development and regeneration, not observed in the other studies.

Globally, our results support a model in which regeneration reuses core developmental programs such as cell type specification and patterning, while also activating unique, regeneration-specific regulatory elements and transcription factors. This is consistent with observations from a recent study in acoels[59]. We therefore favor the interpretation that regeneration only partially reactivates the embryonic program in response to the amputation stress, reflecting the largely differentiated cellular environment of the injury site. Interestingly, this is also confirmed by the findings that genes

associated with the embryonic MEK/ERK (25/88) and cWnt (15/33) GRN modules[26,51] are only partially re-deployed during regeneration.

This transcriptomic comparison between embryonic development and regeneration further revealed a set of genes with expression dynamics specific to the regeneration process (Fig. 1). Subsequent gene-specific analyses will be required to determine the precise role of these genes for regeneration in *N. vectensis* or potentially also in other animals. Our analyses highlighted that most regeneration clusters have significant degrees of conservation with embryonic clusters, while only two regeneration clusters represent no/very reduced conservation (Fig. 2). These observations suggest that embryonic gene modules are re-deployed during regeneration and interconnected to regeneration-specific elements, such as apoptosis and regeneration-specific genes. To experimentally verify the outcome of our global comparative cluster analysis and the prediction that the regeneration GRN follows a reshuffled network logic, we focused on apoptosis, cWnt and MEK/ERK signaling.

This global comparison of genes dynamically expressed during embryonic development and regeneration revealed that apoptosis *N. vectensis* is associated with the regeneration-specific genes and expression cluster R6 (Figs. 1 and 2). These genes belong to so-called pro- (e.g., *bax, caspase-3*) and anti- (e.g., *bcl2, bcl2b*) apoptotic factors[60], highlighting the importance of a fine-balanced regulation of programmed cell death during regeneration. However, the origin of the cells undergoing apoptosis and whether those factors are activated in the same or neighboring cells remains to be elucidated.

Destructive apoptosis has initially been associated with late developmental processes such as digit formation in mammalians[61] and tail regression during metamorphosis in ascidians[62]. Following injury, constructive apoptosis-induced proliferation plays also an important role in tissue repair and regeneration in various metazoans including the freshwater polyp *Hydra*, planarians, *Drosophila*, and Zebrafish (reviewed in ref. 63). Using Z-VAD, a pharmacological pan-caspase inhibitor, classically used to block apoptosis in a variety of research models, we show that apoptosis is a cellular process specific to whole body regeneration in *N. vectensis*, when compared to embryonic development (Fig. 4). It is noteworthy to mention that a recent study in *N. vectensis* has shown that apoptosis is required to eliminate a population of neurons during metamorphosis[54]. Our results show that Z-VAD treatments do not affect metamorphosis, however, treated polyps are smaller in size (Fig. 5A). This suggests a homeostatic role of apoptosis, potentially via the control of neurons that have been shown to be associated with body size regulation in *N. vectensis*[64].

When apoptosis is blocked during regeneration, maintenance of the TC between the mesenteries and the amputation site as well as cell proliferation are prevented. We conclude that not only the classically described "destructive" function, but also the "constructive" function of apoptosis[63] are required to trigger a fine-tuned regenerative

response in *N. vectensis*. In *Hydra*, apoptotic cells are releasing Wnt3 leading to the induction of cellular proliferation at the amputation site[65,66]. Our transcriptomic analysis following Z-VAD treatments did not reveal any specific mitogenic factor that could play a similar role in *N. vectensis*. While our study cannot exclude that a potential mitogenic signal might be diluted by the non-apoptotic cells, our experiment that addressed the functional relationships between these pathways suggest that apoptosis does not activate cWnt, but that these two pathways potentially control each other mutually (Fig. 6).

cWnt signaling has been shown to be a key player in various whole-body regeneration models, including planarians, acoels and cnidarians[67–72], reviewed in ref. 73. In *N. vectensis*, ectopic activation of cWnt following injury can lead to ectopic head formation in aboral regions[33], suggesting that cWnt may play a role in the oral regeneration process. In line with this idea, we showed that inhibition of cWnt following sub-pharyngeal amputation prevents cell proliferation (initiation and maintenance) and progression through the regenerative program.

However, initiation of the *TC* between the *MES* and *BWE* is only slightly delayed, its maintenance is unaffected, and the regeneration process blocked only at the onset of pharynx reformation, i.e., stage 2 (Fig. 3). The expression pattern of *tcf* increases after wound-healing is completed and reaches a peak at 20 hpa. Nuclearization of ß-catenin seems to follow a similar pattern, i.e., is detected at 12 hpa and 24 hpa at the amputation site. This pattern suggests a role of this pathway in the progression of the regeneration process that has been initiated by MEK/ERK. GO terms enriched in genes downregulated following cWnt inhibition during regeneration suggest roles in stem cell division, extracellular matrix, mouth formation and/or interactions with BMP signaling. Some of these biological processes are in line with observed phenotypes, i.e., loss of cell proliferation (this study) or ectopic head formation after cWnt activation[33].

Like a variety of other metazoans, including *Hydra*, planarians, zebrafish, and other vertebrates[74–78] MEK/ERK signaling in *N. vectensis* is also crucial for regeneration (Fig. 5). Furthermore, although a slight delay is observed, we show that MEK/ERK is not required for wound-healing in this animal. Importantly, we found that MEK/ERK signaling is essential for initiating and maintaining the TC between *MES* and *BWE*, initiating and maintaining cell proliferation as well as providing a global input into the regeneration GRN. Further analysis of the upstream activator of MEK/ERK required for *TC*, and the subsequently triggered molecular tissue crosstalk, will contribute to have a better understanding of how this pathway is activated in response to injury to initiate and coordinate a regenerative response.

MEK/ERK signaling has been shown to activate programmed cell death (apoptosis) during *Hydra* regeneration[65,66]. In *N. vectensis*, puncture[38] or sub-pharyngeal amputation (this study) induces apoptosis shortly after injury (Fig. 4D). Inhibiting MEK/ERK signaling in *N. vectensis* after puncture or shortly after head-amputation does not prevent apoptosis[38] (this study, Fig. S17). Surprisingly, we further show that apoptosis is MEK/ERK dependent at 20 hpa (Fig. 6Ac and Supplementary Fig. S17). This suggests that the different waves of apoptosis are regulated differently. In line with this idea, TUNEL+ cells were absent in Z-VAD (pan-caspase inhibitor) treated polyps at 1.5 hpa (Fig. S8A), while still present at 20 hpa (Fig. 6Ab and Supplementary Fig. S17), suggesting that the 2nd wave of apoptosis is caspase-independent. Furthermore, recent work in *N. vectensis* has shown that inhibition of ROS signaling does not affect apoptosis at 2 hpa but is significantly downregulated at 20 hpa[42]. These observations could indicate that the 2nd wave of apoptosis is related to oxeiptosis, a form of regulated cell death operated independently of caspases and induced by ROS[79].

The mechanism activating apoptosis in *N. vectensis* appears to be a hybrid situation between *Hydra* (in which MEK/ERK activates apoptosis[65,66]) and planarians in which an apoptosis-independent role

of MEK/ERK has been described to initiate and maintain the regeneration program[75,76,78]. This could be a consequence of the increased anatomical complexity displayed by *N. vectensis* (anthozoan cnidarian) and planarians (platyhelminth lophotrochozoan) when compared to *Hydra* (hydrozoan cnidarian).

Our data highlight the crucial role of MEK/ERK for the 2nd wave of apoptosis and activation of cWnt during the regeneration process in *N. vectensis*. This is supported by the experiments we performed to address the functional relationships between these pathways (Fig. 6) and transcriptomic data (Fig. 7). In addition to the GO-term enrichment that highlight apoptosis and Wnt signaling in the genes downregulated by U0126, the data show that MEK/ERK controls expression of the apoptosis regulator *bax* and various members of the Wnt pathway (e.g., *wls, wnt4, wnt7*) at 20 hpa (Fig. 7D).

The results from the experiments that addressed the functional relationships between these pathways in combination with the transcriptomic data also reveal that at 20 hpa cWnt and apoptosis seem to mutually control each other. It is therefore interesting to note that the incompletely regenerated polyp observed in the context of inhibiting at least one of the apoptosis waves (Supplementary Fig. S4), and the polyps resulting from cWnt inhibition (Fig. 5), lead to similar phenotypes. This suggests a strong interaction between these two processes/pathways in later steps for differentiation and final progression leading to the reformation of the mouth and tentacle growth.

While MEK/ERK, apoptosis and cWnt share little molecular downstream targets it is interesting to note that all three are required independently for the initiation and maintenance of cell proliferation (Fig. 4D). This interest is increased knowing that TC between *MES* and *BWE* is required for initiating cell proliferation[37] and that the roles of these three pathways on TC is different (Fig. 4C). Thus, these observations suggest different mechanisms involved in controlling initiation of cell proliferation via TC-dependent (MEK/ERK) or TC-independent mechanisms (apoptosis, cWnt). In addition, inhibition of cell proliferation by inhibiting either of the pathways lead to different overall phenotypes (MEK/ERK - stage 0.5; apoptosis - stage 1; cWnt - stage 2) suggesting that proliferation of different cell types/populations may be affected by either of the pathways during the initiation of regeneration in *N. vectensis*.

In the present study, we used a combination of an unbiased large-scale bioinformatics approach and signaling pathway targeted approaches to compare embryonic development and regeneration at the gene network level. Doing so, we suggest and experimentally confirm that the regeneration GRN is a partial and reshuffled redeployment of embryonic GRN modules interconnected with regeneration-specific elements (Fig. 8). In fact, during embryonic development cWnt and MEK/ERK activate a distinct set of downstream targets[26,51], while during regeneration MEK/ERK signaling can activate not only embryonic MEK/ERK targets but also embryonic cWnt as well as "regeneration-specific" genes. Our functional genomics approach thus confirmed the observation resulting from our initial cluster comparison (Fig. 2) suggesting that the GRN underlying regeneration follows a regeneration-specific network logic. This newly established network logic integrates distinct embryonic network modules (MEK/ERK and cWnt) and regeneration-specific elements to rapidly reform lost body parts (Fig. 7).

When looking at these pathways and their roles during embryonic development and regeneration more globally, one can already observe an important molecular reorganization. In fact, during embryonic development, late blastula stages and at the onset of gastrulation, cWnt activates first a set of downstream targets[26] that is largely distinct from the ones controlled in a second phase by MEK/ERK signaling[51]. Although recent work proposes an additional view on the role of cWnt signaling during early embryonic development[30], activity of MEK/ERK during early blastula stages remains segregated to the future mesoderm activating a distinct set of genes compared to cWnt[80]. Apoptosis

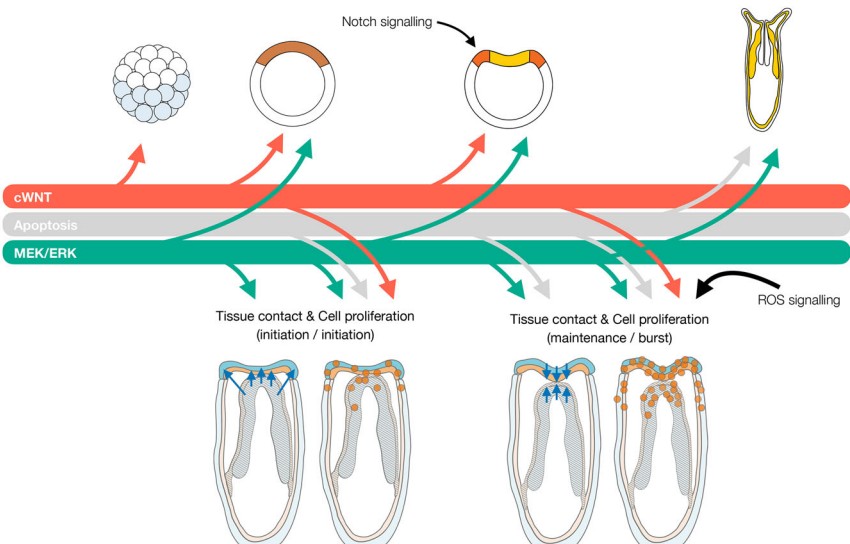

**Fig. 9 | Schematic representation of the roles of the studies pathways during embryonic development and regeneration.** See text for references.

in turn has only been described to be involved in the removal of a subset of neurons during the process of MEK/ERK-dependent metamorphosis[54,56] (Fig. 9).

During regeneration, however, MEK/ERK signaling emerged as the first signaling pathway that plays a critical role with a broad effect. This pathway is activated shortly upon amputation, followed by apoptosis and cWnt signaling. The phenotypes obtained following inhibition of these pathways during wound-healing, *TC*, cell proliferation and regeneration thus highlight the different sequential order of MEK/ERK, apoptosis and cWnt during regeneration, compared to embryonic development.

At this stage, we cannot provide details on the direct or indirect regulatory inputs of the effector of MEK/ERK on these downstream targets, thus further studies looking at chromatin accessibility during regeneration are required. This is particularly important to identify regeneration-specific enhancers[81,82] that may stimulate a regenerative response in tissues that have lost this capacity. Recent studies in the acoel *Hofstenia miamia* has revealed that the transcription factor Egr (Early growth response) is activated early upon injury and controls the expression of a large set of downstream targets, i.e., the Egr-GRN[59,83]. While Egr may play a similar role in planarians[84,85] and potentially in brittle stars[20], its expression pattern during *N. vectensis* regeneration (NvERTx.4.69506) does not support an evolutionarily conserved regeneration-induction function in cnidarians. However, another transcription factor, runx/runt[74,83,86] (this study), might be a conserved key player at the onset of regeneration within metazoans (e.g., Sea star, acoel, *Hydra, N. vectensis*), whose expression in response to injury is regulated by MEK/ERK signaling.

The combination of comparative transcriptional profiling and signaling pathway based functional assays shown in this study, clearly highlights the utility in considering not just individual gene use but how those genes are arranged into co-expression and gene regulatory modules. The identification of expression clusters along with a set of "regeneration-specific" genes provide valuable information to identify, in the future, regeneration-specific genes and enhancers that drive injury-induced expression. Further studies, especially those comparing the activation of the regeneration GRNs across species, will provide insight into our understanding of why certain organisms can regenerate while others cannot. Furthermore, identification of key elements involved in re-deployment of signaling pathways may unlock hidden regenerative potential in poorly regenerating organisms.

## Materials and methods

### Animal culture, spawning, embryo rearing, and amputation
Adult *N. vectensis* were cultured at 16 °C in the dark in 1/3 strength artificial sea water (⅓ASW) as previously described[87]. The day before spawning animals were fed with oysters and were then transferred to a light table for 12 h. Embryos were cultured at 18 °C in the dark in ⅓ASW until desired timepoint. Regeneration experiments were performed using 6 weeks-old juveniles or more than 2-month old sub-adults raised at 22 °C in the dark in ⅓ASW as previously described[36]. Gender was not taken into account in this study as males and females are morphologically indistinguishable at these stages.

### Pharmaceutical drug treatments
Apoptotic cell death was blocked using the pan-caspase inhibitor Z-VAD-FMK (named Z-VAD throughout the manuscript, #ALX-260-020-M001, Enzo Life Sciences Inc, Farmingdale, NY, USA). A stock solution at 10 mM in DMSO was prepared for Z-VAD, kept at −20 °C and diluted in ⅓ASW at a final concentration of 10 µM or 50 µM prior to each experiment (Fig. S4). Each Z-VAD treatment was performed in a final volume of 500 µl in a 24 well plate using the adequate controls. Reagents were changed every 24 h to maintain activity for the duration of the experiments. Phosphorylation/activation of ERK was prevented using the MEK inhibitor U0126 (Sigma-Aldrich U120-1MG). A stock of 10 mM in DMSO was prepared and stored at −20 °C. Each treatment with U0126 were performed with a final concentration of 10 µM in ⅓ASW. Treatments were carried the same way as Z-VAD treatments. cWnt activation was prevented using iCRT14, a β-catenin-responsive transcription inhibitor (Sigma-Aldrich - SML0203-5MG). A stock of 10 mM in DMSO was prepared and stored at −20 °C. Each treatment with iCRT14 was performed with a final concentration of 20 µM in ⅓ASW. Treatments were carried the same way as Z-VAD & U0126 treatments.

### Sample collection and RNA extraction
Sample collection and RNA extraction for the embryonic and regeneration time series are described in refs. 39,88. For pharmacological treatment experiments (biological triplicates), adult animals were first induced to spawn, then amputated below the pharynx and incubated for 20 h in one of the following treatments: 0.1% DMSO (control), 10 µM U0126, 50 µM Z-VAD, or 20 µM iCRT14. At 20 hpa, oral tissues from 10 individuals per condition were dissected and placed into RNAlater (ThermoFisher; #AM7021), transferred to TRI-Reagent

(Sigma-Aldrich; #T9424) in Precellys tubes (Bertin; #CK14), lysed (2 × 0.5 s), flash-frozen in liquid nitrogen, and stored at −80 °C. RNA was purified using a double TRI-Reagent extraction with Phase Lock Gel tubes (QuantaBio; #10847-802) and resuspended in 20 μL of nuclease-free water. Potential DNA contamination was removed with TURBO DNase (2U, 37 °C, 25 min) and inactivated at room temperature (5 min). RNA concentration and integrity were confirmed with an Agilent Bioanalyzer; all samples had RIN values > 8.

## Library preparation and sequencing
cDNA libraries were prepared by the IRCAN Genomed facility using the TruSeq Stranded mRNA Library Prep Kit (Illumina #NP-202-1001) or the Illumina NextSeq 500/550 High-Output v2.5 kit (#20024906). Libraries were sequenced on the Illumina NextSeq 500 platform (75 bp single-end reads).

## Read processing, mapping, and quantification
Raw reads were demultiplexed using bcl2fastq and quality-filtered using Trimmomatic[89], fastp[90], and Cutadapt[91]. Processed single-end reads were aligned to either a combined transcriptome using Bowtie2[92] or to the *genome* (JGI/Nemve1[93]) using STAR[94]. Transcript-level quantification was performed using RSEM[95] for transcriptome-aligned data, and HTSeq[96] for genome-aligned data. Gene-level counts were aggregated by grouping transcripts with the same top BLASTn hit to Nemve1 filtered gene models. Gene models that did not have >5 counts in at least 25% of the samples, were excluded. Normalization was conducted independently for each dataset using the edgeR R package[97]. Count-per-million (CPM) values were calculated, and batch effects across embryonic datasets were corrected using the ComBat function from the SVA package[98], with developmental timepoint as a categorical covariate. For all downstream calculations only embryonic timepoints from 7 hpf (the estimated onset of zygotic transcription) onward are considered so that only gene transcription dynamics are measured and compared.

## Differential expression analysis
Differentially expressed genes (DEGs) were identified using DESeq2[99] for pharmacological treatment samples, with DMSO 0.1% as the reference. Genes with an absolute $log_2FC > 1$ or $>−1$ and an adjusted $p < 0.05$ were considered significant. For time-series datasets (Fischer, Helm, Warner, and regeneration), differential expression was calculated using edgeR by comparing each timepoint to its respective baseline ($t_0$: 7 hpf for Fischer and Helm; 24 hpf for Warner; and 0hpa for regeneration). Genes were classified as significantly differentially expressed if they had an absolute $log_2FC > 1$ or $<−1$ and false discovery rate (FDR) < 0.05. These differentially expressed gene lists were compared to identify overlapping and regeneration specific genes. Venn diagrams to visualize overlapping DEGs across conditions were generated using the VennDiagram R package[100].

## Identification of "regeneration-specific" genes and GO-term analysis
Differentially expressed gene lists were compared to identify overlapping and regeneration specific genes. GO term enrichment were calculated using a Fisher's exact test and the R package topGO on the GO terms identified from comparing the Nemve1 gene models to the UniProt databases Swissprot and Trembl using the BLASTx like program, PLASTx (evalue cutoff 5e-5)[101]. All identified GO terms were used as a background model. The resulting GO term list was reduced and plotted using a modified R script based on REVIGO[102].

## PCA, Fuzzy c-means clustering and cluster conservation
The ensuing analyses were performed using $log_2(cpm+1)$ transformed gene-level quantification (Nemve1 filtered gene models). The expression profiles for each Nemve1 gene model were clustered using the R package mFuzz[103] on the combined embryonic dataset and the regeneration dataset separately. The cluster number was set to 9 for the regeneration data and 8 for the embryonic datasets, as these numbers produced well-separated clusters with minimal overlap (Fig. S3) and represent the inflection point at which the centroid distance between clusters did not significantly decrease with the addition of new clusters (Fig. S3). Genes that did not have a membership score above 0.75 were considered noise and designated as cluster 0. Cluster overlap was calculated for genes that were detectable in both datasets using the function overlapTable from the R package WGCNA, using the regeneration cluster assignments as the reference set. A zStatistic of cluster preservation was also calculated using the function coClustering.permutationTest from the WGCNA package[104] using the regeneration cluster assignments as the reference set and 1000 permutations.

## RT-qPCR
RNA Extraction and RT-qPCR were performed following the protocol from ref. 26. One hundred and sixty juveniles were used per replicate, and each experiment was performed in three biological replicates. We used a 7900HT Fast Real-Time PCR System with 384-Well Block Module (Applied Biosystem) with Faststart universal SYBR Green Master (rox) (FSUSGMMRO Roche). The relative fold change for each gene expression was calculated by comparing treated and untreated conditions, where gapdh or actin was used to normalize fold change (FC).

**Animal fixation.** After relaxing *N. vectensis* polyps in MgCl₂ 7.14% in ⅓ASW for 10–15 min, animals where fixed as followed: (i) for *Whole mount* in situ *hybridization* in cold Glutaraldehyde 0.2% (25% stock Electron Microscopy Sciences; #16216) + paraformaldehyde 4% (32% stock Electron Microscopy Sciences; #15714) in ⅓ASW for 2 to 5 min on ice, then with paraformaldehyde 4% in ⅓ASW for 1 h on ice. Fixed animals were washed five times in PBTw 0.1% (PBS1x + Tween 0.1%) in nuclease free water (NFW), then twice with MeOH 100% and stored at −20 °C; (ii) for *Immunostaining and other stanning* in 4% paraformaldehyde (Electron Microscopy Sciences; #15714) in ⅓ASW during 1 h at 22 °C or overnight at 4 °C. Fixed animals were washed three times in PBT 0.5% (PBS1x + Triton 0.5%) in NFW and stored at 4 °C.

## Whole mount in situ hybridization
Whole mount in situ hybridization was performed as previously described[105]. 0.5–2 kb digox-igenin-labeled (Roche, #11573152910) riboprobes were synthe-sized using the MegaScript Transcription Kit (Ambion). Hybridization of riboprobes (1 ng/ml diluted in fresh hybridization solution) was carried out at 62 °C in 50% formamide hybe buffer and visualization of the labeled probe was performed using NBT/BCIP as substrate for the alkalinephosphatase-conjugated anti-DIG antibody (AP-anti-Dig, Roche,#11093274910). AP/Anti-Dig was used at 1:5000 in blocking solution and incubated at 4 °C overnight. in situ hybridization images were taken on a Zeiss Z-1 Axioimager mounted with an Axiocam camera triggered by the Zen software (Carl Zeiss).

## Immunostaining
The phosphorylated form of ERK was visualized using a monoclonal antibody directed against its active form, pERK (Cell Signaling Technology; #4377) and the protocol used was described in ref. 38. To be able to visualize the nuclearization of ß-catenin::mCherry (i.e., the activated form) we used the primary anti-mCherry antibody (dilution:1/200, Abcam; #ab167453), an Alexa Fluor 488 goat anti-rabbit secondary antibody (dilution:1/200, Invitrogen; #A11008) and followed the protocol described in ref. 106.

## Hoechst and Phalloidin staining

For morphologically characterizing the regeneration step, we counterstained the nuclei and actin filament with Hoechst (Invitrogen; #33342) and BODIPY FL PhallAcidin 488 (Molecular Probes; #B607), respectively. Hoechst was diluted to 1/5000, and BODIPY FL PhallAcidin 488 was diluted to 1/200 in PBS1X.

## EdU staining

For cell proliferation analysis, we used the Click-iT EdU kit (Invitrogen; #C10646, #C10337 or #C10339) following the protocol from ref. 34 with the following change: Juveniles were starved for a week to remove homeostasis proliferation and labeled with 5'-ethynyl-2'-deoxyuridine (EdU 0.1 mM in ⅓ASW) for 1 h, then washed 3 times in ⅓ASW prior fixation and EdU revelation.

## Apoptotic cell death staining

To detect cell death the "In Situ Cell Death AP kit" (Roche; #11684809910) was used. The manufacturer protocol was modified as described as follows: (1) Fixed animals were permeabilized using 0.01 mg/ml Proteinase K for 20 min at 22 °C; (2), Washed twice in PBS1x; (3) Refixed in 4% paraformaldehyde in PBS1x for 1 h at 22 °C; (4) Washed 5 times in PBS1x; (5) Incubated with 50 μL of TUNEL reaction mixture for one hour (Roche protocol); (6) Washed 5 times in PBS1x; (7) Fixed animal were observed for 488 fluorescence. Positive (DNase I treatment after step 4) and negative (without TUNEL-Enzyme) controls were obtained using the manufacturer's "In Situ Cell Death AP kit" protocol.

## Generation of a Nvß-catenin::mCherry KI line

To generate the Nvß-catenin::mCherry KI line, we used a short-homology arm approach described in refs. 107,108 and CRISPR/Cas9 genome editing performed following the protocol described in ref. 109. Briefly, sgRNAs targeting the sequences indicated below were determined using the ZiFiT Targeter website (http://zifit.partners.org/ZiFiT_Cas9) and cloned into the pDR274 vector to create a gRNA expression vector for each target site of interest[110]. sgRNAs were transcribed using the T7 Megascript kit (Invitrogen; #A57622) and cleaned with the Megaclear kit (Invitrogen; #AM1908). To obtain the donor template, mCherry was amplified using primers that contained short mCherry sequences and shorts homology arms (-70 bp). The two sgRNA and repair template were co-injected together with Cas9 into fertilized and dejellied zygotes following the protocol described in ref. 111. Founder F0 polyps expressing the transgene in a mosaic manner were determined visually and raised to sexual maturity and outcrossed to obtain heterozygous F1 (Supplementary Fig. S7B). Offsprings of these F1 were used for experimentation.

sgRNA target sequences
sgRNA1: 5'-CCAGCCAAGTGTTCATC-3'
sRNA3: 5'-GGAGGACACTACCAAAAT-3'
Homology arms (lower case letters indicate mCherry sequences)
FWD_homology_arm:5'CATGGCGACCCGAACATGAGTCATCAGCCACCGGGAGGACACTACCAAAATCCAGGGGGTCCTCTGTATGACACAGACAtggtgagcagggcgagga3' REV_homolgy_arm:5'CTTTTGACGTTGCAATCAACACCCAGCTTTGGTGATGAACACTTGGCTGGCGGCCAAAAGGGGGGGTGTGGATTATttacttgtacagctcgtcca-3'.

Genotyping primers
Fwd1: 5'-CATGGACACGTACCAGATGC-3'
Rev1: 5'-TACAAAATTCGCGAGAACGA-3'

## Protein extraction and western blotting

The protein extracts were obtained from 15 adults per replicate and each experiment was performed in triplicate. The animals were placed in 1.5 ml Eppendorf tubes and spun on bench centrifuge to remove all 1/3 ASW before adding 300 μl of lysis buffer (HEPES 50 mM, NaCl 150 mM, NaF 100 mM, EDTA 10 mM, NA4P207 10 mM). Then animals were sonicated 5 ×10 s with incubation on ice between each round. After sonication 200 μl of Lysis buffer complemented with 1% Triton X-100 and a protease inhibitor cocktail (Apoprotine 20 μg/ml, Vanadate 1 mM, AEBSF 250 μg/ml, Leupeptine 5 μg/ml) was added. The samples were then centrifuged for 20 min at 13,500 × g at 4 °C before collecting the supernatant into new tubes. The BC assay protein quantification kit (Interchim Upima, 40840 A) was used to quantify the protein concentration. Samples were subsequently aliquoted by mixing 75 μl of protein extract with 25 μl of 4X Laemmli buffer (Bio-Rad #1610747) and stored at −20 °C. Electrophoresis was carried out in 7.5% SDS polyacrylamide gel and samples were denatured at 85 °C for 5 min before loading. For each sample, we used 30 μg of protein and followed the protocol described in ref. 18. Briefly, migration was carried in migration buffer (Tris 3 g/L, Glycine 14.2 g/L, SDS 1%) and transfer to the nitrocellulose membrane was performed in transfer buffer (Tris 3 g/L, Glycine 14.4 g/L, 20% Ethanol). Then the nitrocellulose membrane was saturated with Salin buffer (Tris 0.24 g/L, NaCl 1.63 g/L, 5% cBSA, 0.5% Tween-20) and incubated 1 h at room temperature with 1/2500 of primary antibody anti-pERK (Cat.#4377; Cell Signaling Technology) diluted saline buffer. Revelation was carried out by chimioluminescence (EMD Millipore™ Substrats of chimioluminescence HRP Western Luminata™) on a chemioluminescence Imaging −Fusion SL (Vilber).

## TOP FLASH assays

Reporter gene assays were performed based on ref. 112. The TOPFlash reporter plasmid (Addgene; plasmid #12456) was amplified in competent E. coli and purified using standard plasmid extraction methods. The injection mix was prepared by diluting the plasmid to 25 ng/μL in nuclease-free water containing 1× Dextran Green 488 as a tracer. The plasmid solution was microinjected into freshly fertilized and de-jellied N. vectensis embryos. Following injection, embryos were washed and incubated overnight at 17 °C in one of the following conditions: 0.1% DMSO (control), 10 μM U0126, 20 μM iCRT14, or a combination of 10 μM 1-Azakenpaullone (AZ; Sigma Aldrich A3734) and 20 μM iCRT14. Embryos were collected at 24 hpf for analysis. Luciferase activity was measured using the Luciferase Assay System (Promega; #E1500). Embryos (-100 per replicate) were lysed in 100 μL of lysis reagent, and the lysate was centrifuged to collect the supernatant. For each measurement, 20 μL of lysate was combined with 100 μL of assay buffer, and luminescence was immediately recorded using a Centro XS3 LB 960 luminometer (Berthold Technologies) with MikroWin 2000 software. Three biological replicates were performed per condition, each of which was measured in technical triplicates.

**Statistical analysis.** Prior to inferential testing, data distributions were assessed for normality using the Shapiro−Wilk test, and for homogeneity of variances using Bartlett's test. These tests were conducted independently for each dataset and timepoint. For wound-healing assays, since the data were categorical, pairwise Fisher's exact tests were used to compare each experimental condition against the DMSO-treated control at each timepoint. For EdU incorporation data, following normality and variance assessments, pairwise Wilcoxon−Mann−Whitney tests (non-parametric) were applied to compare each treatment to the DMSO control at each timepoint, as data were not normally distributed. For luciferase assay, following normality and variance assessments, Kruskal−Wallis test (non-parametric) was applied. To identify specific group differences, post-hoc comparisons were conducted using Dunn's test with a Benjamini−Hochberg (BH) correction for multiple testing.

All statistical analyses were conducted using R (version 4.3.2). A significance threshold of $p < 0.05$ was applied throughout. Statistical results are presented in Supplementary Data 26. Throughout the manuscript, $n$ = number of biological samples.

## Data availability

The RNAseq data generated in this study have been deposited in the European Genome-Phenome Archive (ENA) with the study accession number PRJEB104129 and data/run accession numbers ranging from ERR15934891 to ERR15934905. All other data generated in this study are provided in the Supplementary Information/Source Data file. All resources associated with this study can be obtained upon request to the corresponding authors. Source data are provided with this paper.

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

## Acknowledgements

We thank Marina Shkreli (IRCAN), Eric Gilson (IRCAN) and Gianni Liti (IRCAN) for suggestions and critical reading of the manuscript; the Röttinger team (past and current members) for fruitful discussions and advice; Renaud Rebillard and Valérie Carlin from the IRCAN ANTIAGE facility for animal husbandry and care. The authors also acknowledge the IRCAN's Genomics Core Facility (Genomed), the Molecular and Cellular Core Imaging (PICMI) Facility and the IRCAN Bio-informatics service. Equipment acquisition for the ANTIAGE facility was supported by Université Côte d'Azur—IDEX UCAJedi (ANR-15-IDEX-01), Région SUD, Canceropôle PACA, Conseil Départemental 06, CNRS Biologie, the ANR RENEW (ANR-20-CE13-014), GIS FC3R (via Inserm) and the GIS Ibisa. Genomed was supported financially by FEDER, Conseil régional Provence Alpes-Côte d'Azur, Conseil départemental 06, Avesian/ITMO Cancer and INSERM. PICMI was supported financially by FEDER, Conseil régional Provence Alpes-Côte d'Azur, Conseil départemental 06, Cancéropôle PACA, Gis Ibisa and INSERM. This work was supported by an ATIP-Avenir award (Institut National de la Santé et de la Recherche & Centre National de Recherche Scientifique) funded by the Plan Cancer (Institut National du Cancer, C13992AS), Seventh Framework Programme (CIG #631665), Fondation ARC pour la Recherche sur le Cancer (PJA2014120186), the French Government (National Research Agency, ANR) through the "Investments for the Future" programs LABEX SIGNALIFE (ANR-11-LABX-0028), IDEX UCAJedi (ANR-15-IDEX-01) and RENEW (ANR-20-CE13-0014) to E.R. K.H. was supported by funding from the European Union's Horizon Europe research and innovation programme under the Marie Skłodowska-Curie Actions (MSCA-DN) grant agreement No 101073238 to E.R. This project has also received funding from the Fondation ARC (PDF20141202150) to J.F.W., Fondation pour la Recherche Médicale to A.R.A. (SPF20130526781), J.E.C. (SPF20170938703) and H.J. (#FDT20170437124), the Ministère de l'Enseignement Supérieur et de la Recherche to R.A.P. and H.J. and the Ligue contre le Cancer to K.N.

## Author contributions

E.R. conceived and designed the study with further contribution from A.A. J.F.W. and R.A.P. performed the comparative and the treatment specific transcriptomic analyses respectively with support from O.C. R.A.P., A.A., and H.J. performed the functional analyses with contributions from K.N., J.E.C., K.H., and A.M. R.A.P., H.J., J.F.W., A.A., and E.R. analyzed the data. ER provided financial support with further contribution from A.A. H.J., J.F.W., A.A., and E.R. wrote the initial manuscript. R.A.P., A.A., and E.R. implemented the new data and revised the manuscript. All authors read and approved the final version of the manuscript.

## Competing interests

The authors declare no competing interests.
