## [Transparent Peer Review file · Nature Communications]

Whole body regeneration deploys a rewired embryonic gene regulatory network logic

Corresponding Author: Dr Eric Röttinger

Version 0:

Reviewer comments:

Reviewer #1

(Remarks to the Author)

The manuscript of Johnston et al. compares the transcriptional program driving *Nematostella* head regeneration to that of its embryonic development. The authors use previously published RNA-Seq data (Warner et al., 2018; Fischer and Smith, 2013; Helm et al., 2013) to show that transcriptional changes during regeneration are relatively small in comparison to the transcriptome dynamics during embryonic development – a finding, which, in my view, is not too surprising. Transcriptome analyses allowed the identification of genes, whose expression was required for the regeneration process. Some of these genes appear to only be involved in the regeneration but not in the embryogenesis program (although some genes on that list are problematic). Johnston et al. suggest, in line with many other studies in different models and using different methods, that *Nematostella* head regeneration re-deploys parts of the embryogenesis program, but with some regeneration-specific features. Importantly, among the genes upregulated during regeneration, the authors find several key regulators of apoptosis and show that apoptosis is required for successful regeneration but not for the embryonic development. Finally, they address the role of MEK/ERK signaling in the regeneration process. The authors show that wound healing does not depend on MEK/ERK signaling but the onset of regeneration does, and characterize the effect of inhibition of MEK/ERK by QPCR and ISH on a number of genes expressed in different regions of the regenerate.

I think that this paper has to be significantly improved to reach the Nature Communications standard. In its present state, it is a descriptive transcriptomic study with two side-stories (functional analyses of apoptosis and MEK/ERK signaling during regeneration), which are not interconnected. However, the manuscript has a clear potential to become excellent if the authors dig deeper into their important functional findings. Since activation of apoptosis seems to be a unique prerequisite for regeneration, I think, this should be the focus of the paper. However, Johnston et al. need to find out what triggers apoptosis and what is being triggered by it.

To that end, I suggest:

- i) comparing transcriptomes of the ZVAD-treated, U0126-treated and control regenerates at stages 0.5 and 1, identifying the overlap among the regulatory genes significantly affected by the treatments, and characterizing their expression during regeneration;
- ii) checking whether apoptosis, MEK/ERK signaling, and expression of b-catenin signaling targets are induced by wounds not causing tissue loss (punctures, incisions), and during foot regeneration;
- iii) checking the effect of pharmacological upregulation and downregulation of b-catenin signaling and MEK/ERK signaling on apoptosis at the onset of regeneration;
- iv) checking the effect of pharmacological upregulation and downregulation of b-catenin and MEK/ERK on cell proliferation at the onset of regeneration;
- v) checking the effect of ZVAD on pERK (immunohistochemistry) and expression of b-catenin target genes (ISH) in amputated animals;
- vi) attempting to rescue the effect of the ZVAD-mediated inhibition of apoptosis on regeneration by activating b-catenin signaling and/or MEK/ERK signaling.

Among all the proposed treatments, only upregulation of MEK/ERK seems not to have been routinely used in *Nematostella*. However, since FGFRb is expressed in the oral region of *Nematostella*, recombinant FGF incubation appears to be a plausible way to achieve upregulation of MEK/ERK.

Once the authors have experimentally analyzed the links and clearly established the regulatory hierarchy between apoptosis, proliferation, MEK/ERK signaling, and Wnt/b-catenin signaling, and identified and characterized the candidate

genes responsible for the start of regeneration, the article can be re-considered for publication at Nature Communications.

In addition to the general suggestions above, I list specific things for the authors to address in the revision (in order of appearance in the text, not in order of importance).

1. Lines 61-62. Johnston et al. write about genes “specifically expressed or required during regeneration” as opposed to “genes specific to embryonic development” and “re-used” genes, and refer to a paper showing the involvement of Sox2 in the regeneration but not in the development of the hair follicle. This is misleading – Sox2 is regeneration-specific only in the hair follicle context, but it is a crucial regulator of the embryonic development.
2. Lines 93-98. Johnston et al. certainly addressed i) and ii). I am not sure about how the “plasticity of the network architecture” (iii) was demonstrated, and I do not think that the paper shows anything with regard to the “rewiring” (iv). If Fig. 7B is meant, I do not find that this paper convincingly proves experimentally that MEK/ERK is upstream of Wnt/b-catenin signaling during regeneration. If the authors refer to lines 344-362, I am not sure I see evidence for rewiring there: most of the embryonic MEK/ERK targets are still MEK/ERK targets during regeneration.
3. Lines 117-122. How exactly are the phases defined? If by transcriptomic changes, there probably should be some metric to it. How do they correspond to the morphological phases of Amiel et al., 2015?
4. Lines 152-154. How exactly do the authors define “regeneration-specific genes”? I did not check all the 124 genes, of course, but I am sure one cannot call Wntless a “regeneration-specific gene”. According to the authors’ own NvERTx database, Wntless is a maternally deposited transcript, and Wntless gene is strongly expressed throughout the development, which is absolutely not surprising, given the importance of Wnt signaling in development and the fact that Wntless is a critical member of the Wnt secretion pathway.
5. Lines 241-243. The finding of the requirement of apoptosis for regeneration but not for normal development is among the most interesting things reported in the paper. Currently, the results of TUNEL on embryos are “data not shown”. Please show and quantify these data and add them to Fig. 4Bii! Please provide quantification data (Fig. 4Bii) in a form of box plots or individual datum points rather than a table with means, so that it is possible to estimate the variance between polyyps. I also suggest increasing N to at least 5 in every case.
6. Lines 259-260. The authors suggest that apoptosis induces cell proliferation. In order to make the argument that apoptosis is upstream of proliferation more convincing, I suggest directly blocking proliferation (with cyclin-dependent kinase or DNA polymerase inhibitors) and checking whether apoptosis is still detectable.
7. Line 270. The authors have to refer to DuBuc et al., 2014 here. That paper already showed that U0126 blocks head regeneration.
8. Lines 289-290 and Fig. 5F. Is the delay in wound healing significant? The sample size is low, so making biological replicates and identifying the confidence interval is important to understand whether there is a delay at all.
9. Line 295. Can heads re-commence regeneration after U0126 washout? If not, can re-amputation re-start regeneration? Is it possible to create a “dormant Nematostella head” as in <https://www.nature.com/articles/s41467-017-02338-x>
10. Lines 325-327. How did the authors come up with 40 embryonic and 40 regeneration genes? Were these the top 40? Please explain.
11. Lines 327-330. The authors write “From the 40 embryonic GRN genes, 25 have been described as downstream targets of MEK/ERK signalling, while the expression of 15 are controlled by the canonical Wnt (cWnt) pathway at the onset of gastrulation.” Can such a clear distinction really be made? There must be an overlap between the 25 and the 15. For example, WntA and Sp8/9 are clearly not only MEK/ERK but also Wnt/b-catenin targets at gastrulation.
12. Lines 331-342. I fail to see the message here. These genes are co-expressed, so, indeed, they are possibly doing something together. Please rephrase for more clarity.
13. Lines 347-349. Johnston et al. write, “Strikingly, 19 out of the 25 identified embryonic MEK/ERK downstream targets (Table S5) are downregulated by U0126 treatments during regeneration (Fig. 6Bi)”. Why is it striking? I would rather say, “as expected”.
14. Lines 370-372. I do not doubt the involvement of Wnt and MEK/ERK signaling in the regulation of regeneration but I do not see any experimental confirmation of the placement of Wnt downstream of MEK/ERK. This study lacks systematic testing of what happens to regeneration upon upregulation and downregulation of Wnt as well as upregulation and downregulation of MEK/ERK as well as data showing which of the two (or maybe both) pathways is activated by the early wave of apoptosis. Please also do not forget to refer to the Hydra paper of Chera et al., 2009 when talking about the effect of apoptosis on Wnt.
15. Line 396. Remove “and”
16. Line 427. Why does the logic have to be reshuffled? What is the basis for this prediction?
17. Line 429. Remove “and” from “other and vertebrates”.
18. Lines 460-463. Johnston et al. write that “cWNT and MEK/ERK activate a distinct set of downstream targets, while during regeneration MEK/ERK signalling is able to activate not only embryonic MEK/ERK targets but also embryonic cWNT as well as “regeneration-specific” genes.” This is a “false dichotomy”. There is a clear overlap between cWNT and MEK/ERK targets in the embryo (WntA, Sp8/9 – see Kraus et al., 2016 and Lebedeva et al., 2017). Please correct.
19. Line 676. Probably the title of the figure should be “ERK phosphorylation is required for regeneration”.
20. Figure 7A. The authors call this a network module, but it is just a list of MEK/ERK target genes. It will become a network module once the regulatory relationships between the targets are experimentally established.
21. Figure 7B. This scheme is very suggestive but I do not think there is experimental evidence for it. Has it been shown that b-catenin signaling is required for MEK/ERK signaling in the embryos? I do not think there is a publication showing that b-catenin knockdown leads to the absence of MEK/ERK signaling in the embryos. Similarly, this study did not demonstrate experimentally that MEK/ERK is upstream of Wnt/b-catenin. If this is true, b-catenin target genes should be suppressed upon U0126 treatment, but this has to be checked first.

(Remarks to the Author)

The manuscript by Warner et al uses a re-analysis of prior transcriptomic data to address the similarities and differences between embryogenesis and regeneration in the model sea anemone *Nematostella vectensis*. The fundamental question of whether regeneration is simply a redeployment of development has been extensively posed in the literature in prior work. Hence the novelty of this study is primarily restricted to the organismal system and the systems biology methods employed to identify differentially expressed genes. As it appears this part of the work largely draws on analysis of published data, the key contribution amounts to the validation and functional experiments and therefore the impact of the work is fairly field-specific. In that regard, this manuscript provides an interesting starting point but falls significantly below the level of general insight expected for the journal. The experimental validation sections of the paper, which address roles for apoptosis and MEK/ERK signaling in regeneration are a big strength of the paper and represent an important advance in the field.

On the whole, the figures presenting the transcriptomic analysis are significantly below journal standards and the data do not clearly support conclusions drawn by the authors. Several highly general statements e.g. "This indicates the transcriptional dynamics of embryogenesis are more profound than those of regeneration" are so vague as to be meaningless as the analysis of bulk RNA-seq data likely obscures the transcriptional dynamics of critical cells at the wound site that contribute to regeneration. In this regard, a much higher resolution understanding could be gained by scRNA-seq or spatial transcriptomic approach. It is not clear whether or how this part of the work will have impact or utility to the broader community beyond serving as a starting point for the subsequent experiments.

The ZVAD experiments to inhibit apoptosis reveal an interesting requirement for apoptotic cell death in the initiation of regeneration and reflect one of the stronger logical components of the paper. It would be interesting to add an analysis of cell death following wounding without amputation to determine whether cell death can be seen as a consequence of wounding or reflects a component of the regenerative program proper. The TUNEL analysis suggests there may be wound-site specific and systemic cell death upon amputation and the authors could significantly expand the impact of the work by digging deeper in this area.

Similar to the analysis of cell death, the results on MEK/ERK could provide key new insights into *Nematostella* regeneration. Combined with the work on apoptosis, this is the strongest part of the paper. However, it remains unclear how direct or indirect the activation of downstream targets may be and how specifically U0126 treatment affects MEK/ERK signaling. Given the proliferation of genetic methodologies for *Nematostella*, are there not more rigorous approaches available to demonstrate specific gene requirements by CRISPR or shRNA knockdown?

Reviewer #3

(Remarks to the Author)

The manuscript by Johnston et al. uses a comparative approach to determine the relationship between embryonic development and regeneration at both molecular and cellular levels. While defining the key factors that distinguish regeneration from development is fundamentally important, experimentally, only a few research organisms offer the possibility to address this question. In this context, the authors leveraged the accessibility of embryonic development and regeneration in the sea anemone *Nematostella vectensis* and have generated and analyzed impressive transcriptomic data covering the temporal dynamics of both embryogenesis and regeneration. Building on this comparative approach and functional data involving ERK signaling, the authors identified and validated gene expression patterns that reflect a rewiring of the embryonic genetic program. At the cellular level, apoptosis appears to be required for regeneration, which emphasizes the role of this cellular process as a core mechanism deployed across different regenerative species. The great strength of this study is that the authors have produced significant transcriptomic resources for the scientific community, and the advancements presented here certainly establish this sea anemone as an attractive research organism to study the developmental origins of regeneration. In general, the experiments are well performed throughout the manuscript. However, the interpretations of some results would require clarification or additional controls to address the following concerns:

Major concerns:

- 1- The authors claim that "Regeneration is transcriptionally modest compared to embryonic development". However, this conclusion is based on comparing oral pole regeneration, which does not represent the full regenerative capacity of the polyp while embryogenesis builds a whole organism. The authors should reconsider this statement as the current data do not support it.
- 2- An experimental validation of 'regeneration specific' genes is missing. In this context, an attractive category is the list of genes that are taxonomically restricted. This work has the potential to inform us about their integration in the gene regulatory logic. For instance, Is there any bias in their spatio-temporal deployment during regeneration?
- 3- The authors also mentioned that "Apoptosis is specifically required for regeneration" without providing strong evidence to support this claim. Many studies show embryos or larvae might look morphologically normal while cell differentiation can be dramatically impacted when using molecular profiling. In addition, a peak of cell death has been reported during late larval development (Zang and Nakanishi, 2020). An obvious experiment would be to treat late larvae with the pan-caspase inhibitor Z-VAD and test whether development proceeds normally.
- 4- It is exciting to see the involvement of apoptosis in *Nematostella* regeneration. Could the authors explore the

transcriptomic data and show the expression of developmental signals that are provided by these dying cells.

Minor concerns:

- The labeling of the panels with multiple letters is not simple to follow and creates confusion.
- There is a typo in "Apoptosis is a and regeneration-specific process."

Version 1:

Reviewer comments:

Reviewer #1

(Remarks to the Author)

The authors have significantly improved the paper, and experimentally it is now a very solid work. There are several specific things, which, in my opinion, need to be done for the manuscript to become publishable (see the point-by-point below). I also suggest that the authors shorten the discussion making it less speculative and focusing it on the implications of their main findings rather than on what other questions would be potentially interesting to address.

Point-by-point comments in order of appearance in the text (not in the order of importance!).

Line 163. "Regeneration-specific" genes may be regeneration-specific only in this particular comparison. For example, *wntless* is a maternal transcript abundantly present throughout development in the pharyngeal and body wall ectoderm.

Line 333. Fig. S8: In contrast to the staining on Fig. 4L, where nuclear b-cat-mCherry is certainly real at 12 and 24 hpa, the authors misinterpret the b-cat-mCherry staining on Fig. S8D (by the way, D is present in the legend but missing on the figure!). The small circles in the middle of the cells, which the arrowheads are pointing at are not nuclei. They are crowns of microvilli surrounding the cilium, - a similar picture is shown in Fig. 1 of the Momose et al., 2012 on *Clytia*, and on your own Fig 4L in the 48 hpa image. Nuclei are much bigger - see DAPI. Also, the authors show significant depletion of the mCherry signal in the cell contacts in iCRT14 in comparison with DMSO, which should not happen, since iCRT14 prevents b-cat/Tcf interaction, not something else.

Another point: I am not sure what "a", "b", "abc", "ac" and "d" mean on top of the error bars on (C), but to me this plot shows that AZ is the only chemical with a clear effect on b-cat signaling. iCRT14 does not have a significant effect on b-cat in comparison to DMSO. It is not clear to me, how iCRT14 can abrogate the effect of AZ if it does not have an effect on topflash-driven expression when administered alone. A good control for the functionality of iCRT14 would be qPCR or RNA-Seq on b-cat target genes during regeneration. It should be reciprocal in iCRT14 and AZ - at least for genes upregulated in AZ.

Lines 334-335: Clumsy sentence. MEK/ERK and cWnt do not block development and patterning. Experimental modulation of MEK/ERK and cWnt does.

Lines 352-353: The authors write "iCRT14 treated polyps did not display β -catenin nuclearization at 24hpa (Fig.S8C)" – but their controls also do not show b-cat nuclearization - at least on Fig. S8. See comment to Line 333. There is clear nuclear b-cat-mCherry signal on Fig. 4L at 12 and 24hpa.

Line 363: Replace "as" with "in comparison to"

Line 445: On Fig. 7C, the GO-terms are completely unreadable without a 5x zoom, and certainly not on the printout. If this is a main text figure, every part of it has to be readable on a standard page without zooming.

Lines 451-461: How do the authors explain that MAPK signaling is upstream of Wnt, but the genes controlled by Wnt are not affected by MAPK inhibition?

Lines 476-481: The authors write "Systematically comparing these pools of genes (*emb_cWnt*, *emb_ERK*, *regen_spe*) with the up- or down-regulated genes following inhibition of MEK/ERK (Tables S20,S21), apoptosis (Tables S22,S23), or cWnt (Tables S24,S25) during regeneration, we observed that MEK/ERK signaling during regeneration has the most widely extended control on gene expression for all three pools (Fig.8; Fig.S15)." This makes total sense, but doesn't it contradict what they wrote on lines 451-461 (that target lists barely overlap)?

Lines 495-502: I do not find labels like Bi, Bii, and Biii on Fig. 8. I also do not see any insets on Fig. 7B.

Line 514: This is not a GRN. Unlike anything published by Eric Davidson, who was using the same type of graphics for GRN representation, here, it is just a list of downstream targets of MEK/ERK without any experimentally proven or even hypothetical regulatory links between them. It occupies figure space without providing any information a supplementary table cannot provide. Please remove.

Line 534-536: The authors write that their results "support a model in which regeneration reuses core developmental programs such as cell type specification and patterning, while also activating unique, regeneration-specific regulatory elements and transcription factors". They may be right in principle, but their results do not show anything of this sort. Out of

123 “regeneration-specific genes” in Table S1, there is a single transcription factor, Msx2-like, however, single-cell transcriptomics atlas of Cole et al., 2024 shows that it is strongly expressed in the retractor muscle and also in one particular neuroglandular cell type in a normal adult polyp. So, it may be “regeneration-specific” only in the comparison made by Andreoni-Pham et al., but not otherwise. Since there is no ATAC-Seq data in the manuscript, the authors cannot say that their data supports the existence of “regeneration-specific regulatory elements” either.

Line 657: wnt7

Line 703: The scheme on Fig. 9 is incorrect. MEK/ERK is responsible for the initiation of the endomesodermal expression in the plate in the 6-8 hpf blastula stage (Haillot et al., 2025); that is the same time when maternal beta-catenin goes into the nuclei of all future ectodermal cells (Lebedeva et al., 2025). So, there has to be not only the orange but also the green arrows to each embryo cartoon.

Reviewer #2

(Remarks to the Author)

The authors have incorporated significant new functional experiments into the manuscript in F5 and F6. This addresses one major concern. Still, their rebuttal to my comments offered mostly argumentation and minimal specific changes in response to the original comments (excepting one changed sentence and the analysis of the waves of apoptosis in SF9). On the whole, the paper is improved and scientifically sound but I maintain some concerns about the generality of the bulk RNA-seq analysis, the lack of single cell resolution, and thus the general impact of the work relative to the current literature.

Reviewer #3

(Remarks to the Author)

The revised manuscript by Andreoni-Pham, Johnston & Warner et al. represents a substantial improvement over the initial submission. The authors have effectively addressed the major concerns by incorporating additional experiments that strengthen the connections between their transcriptomic analyses and claims. The manuscript now presents a more cohesive narrative that integrates descriptive and functional approaches. I believe the work is now suitable for publication in Nature Communications.

REVIEWER COMMENTS

Reviewer #1 (Remarks to the Author):

The manuscript of Johnston et al. compares the transcriptional program driving *Nematostella* head regeneration to that of its embryonic development. The authors use previously published RNA-Seq data (Warner et al., 2018; Fischer and Smith, 2013; Helm et al., 2013) to show that transcriptional changes during regeneration are relatively small in comparison to the transcriptome dynamics during embryonic development – a finding, which, in my view, is not too surprising. Transcriptome analyses allowed the identification of genes, whose expression was required for the regeneration process. Some of these genes appear to only be involved in the regeneration but not in the embryogenesis program (although some genes on that list are problematic). Johnston et al. suggest, in line with many other studies in different models and using different methods, that *Nematostella* head regeneration re-deploys parts of the embryogenesis program, but with some regeneration-specific features. Importantly, among the genes upregulated during regeneration, the authors find several key regulators of apoptosis and show that apoptosis is required for successful regeneration but not for the embryonic development. Finally, they address the role of MEK/ERK signaling in the regeneration process. The authors show that wound healing does not depend on MEK/ERK signaling but the onset of regeneration does, and characterize the effect of inhibition of MEK/ERK by QPCR and ISH on a number of genes expressed in different regions of the regenerate.

I think that this paper has to be significantly improved to reach the Nature Communications standard. In its present state, it is a descriptive transcriptomic study with two side-stories (functional analyses of apoptosis and MEK/ERK signaling during regeneration), which are not interconnected. However, the manuscript has a clear potential to become excellent if the authors dig deeper into their important functional findings. Since activation of apoptosis seems to be a unique prerequisite for regeneration, I think, this should be the focus of the paper. However, Johnston et al. need to find out what triggers apoptosis and what is being triggered by it.

We thank the reviewer for the relevant comments that lead to a series of additional experiments and the re-organization of our study to better interconnect the transcriptomic aspects with the functional data. We're convinced that these additional data have helped to significantly improve the manuscript for publication in Nature Communication.

To address the connection between apoptosis and MEK/ERK signaling, we have performed additional experiments and show that while apoptosis seems MEK/ERK-independent at 2hpa (Fig. S17), it is dependent of MEK/ERK activity at 20h (Fig. 6 and Fig. S17). Our new data further suggest that the first wave of apoptosis (2hpa) is caspase-dependent (Fig.S8 and Fig. S17) but MEK/ERK independent, while the second wave (20hpa) is caspase-independent (Fig.6b and Fig. S17) but MEK/ERK dependent. This is detailed in the revised discussion and put in relation to recent findings that apoptosis at 20hpa is ROS dependent, while at 2hpa is it not (Vullien et al. 2024). In addition to showing that apoptosis is crucial for regeneration, we now also show that inhibition of apoptosis does not affect wound-healing or the initiation of the tissue-contact between the mesenteries and the body wall epithelia but is required for its

maintenance as well as for both, the initiation and maintenance of cell proliferation (Fig. 5). Finally, we have performed a transcriptomic analysis following apoptosis that has been used in the context of the study, i.e., comparing it to the transcriptomes obtained after inhibition of MEK/ERK and cWNT signaling, as well as with the genes belonging to embryonic MEK/ERK and cWNT downstream targets or the set of regeneration-specific genes identified in our study (Fig.7,8).

To that end, I suggest:

i) comparing transcriptomes of the ZVAD-treated, U0126-treated and control regenerates at stages 0.5 and 1, identifying the overlap among the regulatory genes significantly affected by the treatments, and characterizing their expression during regeneration;

We have performed transcriptomic analyses on 20hpa treated polyps with DMSO (controls), Z-VAD, U0126 or iCRT14 (iCRT14 blocks cWNT signaling) (Fig. 7, S13, S14, and associated supplemental Tables). The overlap between the significantly DOWN or UP-regulated genes shared between all treatments is rather low (5 for the DOWN and 10 for the UP). Their identity and temporal expression pattern that span the entire regeneration process are shown in Fig. 7D. Go term analyses and assessments of the temporal expression of the DOWN- and UP-regulated genes by the various inhibitory treatments confirm the results from the experiments that assessed the relationship between the three pathways (Fig.6), *i.e.*, MEK/ERK controls downstream targets associated to Wnt signaling and apoptosis. Additional information supporting our statement, are indicated in the revised manuscript. Looking globally at the temporal expression patterns (Fig.S14), we observed that MEK/ERK downstream targets seem upregulated during regeneration in three phases, i) in response to injury, ii) at the onset of regeneration (14-36hpa) and iii) during differentiation (96-144hpa). On the other hand, downstream targets of apoptosis and cWNT are mainly activated either shortly after injury or after 36hpa (Fig.S14).

ii) checking whether apoptosis, MEK/ERK signaling, and expression of b-catenin signaling targets are induced by wounds not causing tissue loss (punctures, incisions), and during foot regeneration;

We agree that the suggested experiments to compare the amputation/regenerative response with the injury/wound-healing response or aboral regeneration are certainly interesting, however, in our opinion out of scope of the present manuscript. In fact, the present study focuses on the GRN underlying oral regeneration, and its comparison to the GRN underlying embryonic development. We hope that all the additional experiments, that increased our understanding of the main question of the study, nonetheless satisfy the reviewer.

iii) checking the effect of pharmacological upregulation and downregulation of b-catenin signaling and MEK/ERK signaling on apoptosis at the onset of regeneration;

To assess the activation pattern of cWNT signaling, we have developed a CRISPR/Cas9 induced KI line by fusing mCherry to the C-terminal of β -catenin (Fig. 4L, S7) and observed the beginning of β -cat nuclearization in cells at the amputation site at 12hpa and 24hpa. Using

the readouts for the activation of all pathways (pERK for MEK/ERK signaling, TUNEL for apoptosis, and β -catenin::mCHERRY for cWNT) we have performed additional experiments in line with the focus of our study. In particular, we performed experiments to understand the relationship (*i.e.*, necessity) between the three pathways using inhibitor treatments (Fig. 6). Doing so we were able to show that MEK/ERK signaling is required for apoptosis and cWNT signaling. This is in line with the observation that inhibition of MEK/ERK signaling causes the most severe phenotype by blocking regeneration at stage 0.5 (Fig. 5). While Z-VAD and iCRT14 treatments have no visible effect on pERK activation, our results suggest a mutual regulation between apoptosis and cWNT signaling at the amputation site at 24hpa (Fig. S12).

iv) checking the effect of pharmacological upregulation and downregulation of b-catenin and MEK/ERK on cell proliferation at the onset of regeneration;

As our goal is to determine the necessity of the pathways for regeneration, and not an eventual sufficiency (this is another ongoing study in the lab), we have focused this series of experiments only on the downregulation of the pathways. We have assessed the effects of inhibiting MEK/ERK, apoptosis or cWNT on several and complementary readouts specific to the regeneration process such as wound healing, the onset and maintenance of the tissue contact required for the initiation of regeneration, as well as the onset and maintenance of cell proliferation. All the results are shown now in Fig 5. Our results show that all three pathways are required for injury-induced cell proliferation (Fig. 5D,E).

v) checking the effect of ZVAD on pERK (immunohistochemistry) and expression of b-catenin target genes (ISH) in amputated animals;

We have performed pERK staining following Z-VAD treatments at 20hpa for the above-described experiments (Fig. 6) and show that apoptosis is not required for pERK activation. However, the opposite is the case as TUNEL staining is reduced in UO126 conditions (Fig. 6 and Fig.S17). Regarding the expression of b-catenin target genes in amputated animals, we have obtained a list of potential cWNT downstream target genes during regeneration from our transcriptomics analyses (Fig. 7). We assessed and show their temporal expression pattern during regeneration (Fig. S14). ISH of embryonic cWNT downstream targets during regeneration are shown in Fig. 8C.

vi) attempting to rescue the effect of the ZVAD-mediated inhibition of apoptosis on regeneration by activating b-catenin signaling and/or MEK/ERK signaling.

We agree on the interest of these experiments that further address the tissue competence for the impressive regenerative capacity of *N. vecten**is*. They appear to us out of the scope of the current manuscript and are part of another project from the lab. Following the addition of a wealth of new data characterizing the role of these three pathways during regeneration, we have decided to remain focused on the main question of the manuscript – namely, to what extent regeneration recapitulates the embryonic GRN.

Among all the proposed treatments, only upregulation of MEK/ERK seems not to have been routinely used in *Nematostella*. However, since FGFRb is expressed in the oral region of

Nematostella, recombinant FGF incubation appears to be a plausible way to achieve upregulation of MEK/ERK.

In another study from the lab focusing on the role of the tissue contact and subsequent tissue-crosstalk (Andreoni et al., close to submission), we have a series of evidence suggesting that FGFRb is indeed involved in the injury-induced activation of MEK/ERK and its down-stream targets. However, FGFRb does not control the entire set of downstream targets and other pathways seem to converge in MEK/ERK activation at the onset of regeneration in *N. vectenis*. Preliminary data for another study from the lab, seems to suggest that recombinant FGF signaling alone is not sufficient to induce regeneration.

Once the authors have experimentally analyzed the links and clearly established the regulatory hierarchy between apoptosis, proliferation, MEK/ERK signaling, and Wnt/b-catenin signaling, and identified and characterized the candidate genes responsible for the start of regeneration, the article can be re-considered for publication at Nature Communications.

We thank the reviewer again for the constructive comments. We hope that the additional data sets and detailed analyses of the three pathways during regeneration provide compelling evidence that MEK/ERK signaling is the earliest and most crucial injury-induce signal that triggers an apoptosis and cWNT dependent regenerative response.

Based on our initial data and the new ones we acquired during the revisions, we propose that all three pathways are required for regeneration, the initiation and maintenance of cell proliferation. We further show that MEK/ERK signaling has the most drastic effect in launching the regeneration process (Fig. 5). In line with this idea, we show that MEK/ERK signaling is required for both apoptosis and cWNT signaling at the onset of regeneration (20hpa). Furthermore, our results, in particular the data from Fig. 6 (that specifically addresses the relationships) and the RNAseq analysis (Fig. 7) also support more subtle and non-hierarchical relationships, *i.e.*, the mutual control of apoptosis and cWNT signaling. Importantly, our results enable us to perform a compelling comparison between the deployment of these pathways during embryonic development and regeneration highlighting a reshuffled hierarchy (Fig. 9).

In addition to the general suggestions above, I list specific things for the authors to address in the revision (in order of appearance in the text, not in order of importance).

1. Lines 61-62. Johnston et al. write about genes “specifically expressed or required during regeneration” as opposed to “genes specific to embryonic development” and “re-used” genes, and refer to a paper showing the involvement of Sox2 in the regeneration but not in the development of the hair follicle. This is misleading – Sox2 is regeneration-specific only in the hair follicle context, but it is a crucial regulator of the embryonic development.

We have added “initial development” to the manuscript.

2. Lines 93-98. Johnston et al. certainly addressed i) and ii). I am not sure about how the “plasticity of the network architecture” (iii) was demonstrated, and I do not think that the paper shows anything with regard to the “rewiring” (iv). If Fig. 7B is meant, I do not find that

this paper convincingly proves experimentally that MEK/ERK is upstream of Wnt/b-catenin signaling during regeneration. If the authors refer to lines 344-362, I am not sure I see evidence for rewiring there: most of the embryonic MEK/ERK targets are still MEK/ERK targets during regeneration.

We are convinced that the additional sets of experiments we performed for the revision of this manuscript have largely addressed the concerns raised by the reviewer, in particular that MEK/ERK is upstream of cWNT signaling. The term “rewiring” is mainly linked to two observations we made: A) the simplified temporal deployment of cWNT → MEK/ERK → Apoptosis during embryonic development is not conserved during regeneration but reshuffled with the following temporal deployment of MEK/ERK → Apoptosis → cWNT during regeneration. B) During embryonic development, cWNT activates a largely distinct set of downstream targets than MEK/ERK; During regeneration, these distinct set of downstream targets are controlled by MEK/ERK. More specifically, while most (not all) embryonic MEK/ERK targets are still MEK/ERK downstream targets during regeneration, a subset of embryonic cWNT downstream targets is controlled by MEK/ERK but not anymore by cWNT during regeneration (Fig. 8A). Thus, and because “regeneration-specific” genes are also controlled by MEK/ERK signaling during regeneration, we propose that regeneration redeploys elements of the embryonic gene regulatory network but in a different temporal sequence and rewired / interconnected with “regeneration-specific” genes.

3. Lines 117-122. How exactly are the phases defined? If by transcriptomic changes, there probably should be some metric to it. How do they correspond to the morphological phases of Amiel et al., 2015?

To clarify this point, we added supplementary information to this section.

4. Lines 152-154. How exactly do the authors define “regeneration-specific genes”? I did not check all the 124 genes, of course, but I am sure one cannot call Wntless a “regeneration-specific gene”. According to the authors’ own NvERTx database, Wntless is a maternally deposited transcript, and Wntless gene is strongly expressed throughout the development, which is absolutely not surprising, given the importance of Wnt signaling in development and the fact that Wntless is a critical member of the Wnt secretion pathway.

The reviewer is correct and that is why we defined the “regeneration-specific” genes not because they are only expressed during regeneration or that their roles are only associated to regeneration, but that their expression is dynamic during regeneration (fC > 2-fold) and not during embryonic development (fC < 2-fold). While we have amended the first phrase of this section to make this clearer, the text referring to Figure 1F stated, and still states, that a part of the “regeneration-specific” genes is expressed during embryonic development.

5. Lines 241-243. The finding of the requirement of apoptosis for regeneration but not for normal development is among the most interesting things reported in the paper. Currently, the results of TUNEL on embryos are “data not shown”. Please show and quantify these data and add them to Fig. 4Bii!

We have added the experiment that shows that apoptosis is required for regeneration and not for embryonic development to Fig. 5A. We now also cite the manuscript that showed that TUNEL positive cells appear in the larvae during metamorphosis (Zang & Nakanishi, 2020).

Please provide quantification data (Fig. 4Bii) in a form of box plots or individual datum points rather than a table with means, so that it is possible to estimate the variance between polyps. I also suggest increasing N to at least 5 in every case.

Done.

6. Lines 259-260. The authors suggest that apoptosis induces cell proliferation. In order to make the argument that apoptosis is upstream of proliferation more convincing, I suggest directly blocking proliferation (with cyclin-dependent kinase or DNA polymerase inhibitors) and checking whether apoptosis is still detectable.

When blocking directly proliferation using Hydroxy Urea, regeneration is blocked at stage 1 (Amiel et al. 2015). However, we do not understand how directly blocking proliferation will inform us if apoptosis is upstream of cell proliferation. From our understanding, such experiment will only suggest a potential impact of cell proliferation on apoptosis and not the contrary. By using the pan-caspase inhibitor Z-VAD, we showed that the inhibition of apoptosis blocks not only the initiation but also the maintenance of injury-induced cell proliferation during regeneration (Fig.5D). This shows that apoptosis is upstream to and required for cell proliferation.

7. Line 270. The authors have to refer to DuBuc et al., 2014 here. That paper already showed that U0126 blocks head regeneration.

We have added Dubuc et al 2014 (that we cite elsewhere multiple times and that mainly focused on wound-healing) to the introduction, mentioning that this study suggests that MEK/ERK signaling is required for oral regeneration (without showing the specific numbers that support their observation).

8. Lines 289-290 and Fig. 5F. Is the delay in wound healing significant? The sample size is low, so making biological replicates and identifying the confidence interval is important to understand whether there is a delay at all.

The wound-healing assay on U0126 treated regenerating polyps has been performed in replicates that have been pooled/integrated in Figure 5B (indicating the number of assessed polyps). This figure also shows our assessment of the effects of blocking apoptosis and cWNT signaling on wound healing. Statistical analyses confirmed the delay in wound healing for U0126 treated polyps, while iCRT14 and Z-VAD slightly accelerated the process.

9. Line 295. Can heads re-commence regeneration after U0126 washout? If not, can re-amputation re-start regeneration? Is it possible to create a “dormant Nematostella head” as in <https://www.nature.com/articles/s41467-017-02338-x>

While this is indeed interesting, we consider these experiments out of the scope of the present manuscript and are part of another ongoing project in the laboratory.

10. Lines 325-327. How did the authors come up with 40 embryonic and 40 regeneration genes? Were these the top 40? Please explain.

We have rephrased this section in the manuscript and hope this is clearer now.

To identify the 40 embryonic and the 40 regeneration genes, we have taken the list of the embryonic i) MEK/ERK or ii) cWNT downstream targets identified in Röttinger et al, 2012 and Amiel et al. 2017 respectively, as well as iii) the “regeneration-specific genes” we identified in this current study. Then, we have cross-referenced these three gene lists with the differentially expressed genes at 20hpa. Doing so we have identified the mentioned 40 genes associated to embryonic MEK/ERK and cWnt downstream targets as well as 44 “regeneration-specific” genes (we updated this number in the manuscript, Fig. 3A,B; S3).

11. Lines 327-330. The authors write “From the 40 embryonic GRN genes, 25 have been described as downstream targets of MEK/ERK, while the expression of 15 are controlled by the canonical Wnt (cWnt) pathway at the onset of gastrulation.” Can such a clear distinction really be made? There must be an overlap between the 25 and the 15. For example, *WntA* and *Sp8/9* are clearly not only MEK/ERK but also Wnt/b-catenin targets at gastrulation.

Based on the data from the cited papers (Röttinger et al 2012 and Amiel et al, 2017) our statement is largely correct. Figure 7 from Amiel et al, 2017 shows the downstream targets of MEK/ERK and cWnt signaling during the onset of gastrulation. We have attentively compared the pathway specific downstream targets (Table S5, S6) and the only genes that indicate an input from cWnt and MEK/ERK during embryonic development are *nfix-like*, *hd050*, *elk-like1* and *gsc*. If the reviewer can send us the references that show that *wntA* and *sp8/9* are both controlled by MEK/ERK and cWnt at the onset of gastrulation, we would be happy to include these references.

We have re-written this section and to consider that not strictly all genes are downstream of one or the other pathway, we reformulated the sentence in the rewritten paragraph. “These embryonic downstream targets are largely pathway specific as only 4 genes (*nfix-like*, *hd050*, *elk-like1* and *gsc*) are controlled by MEK/ERK and cWNT at the onset of gastrulation (Table S5,6).”

12. Lines 331-342. I fail to see the message here. These genes are co-expressed, so, indeed, they are possibly doing something together. Please rephrase for more clarity.

We make a distinction between “co-expression” and “syn-expression”. Co-expression can be temporal (*i.e.*, the expression patterns shown in Figure S2) or spatial (*i.e.*, some of the expression pattern shown in Figure 5). Syn-expression, as defined by Niehrs et al., 2000 (that we now cite for additional information) are genes that are expressed identically at both level spatial and temporal. In short, genes that belong to syn-expression groups are more likely to participate in the GRN underlying a common biological process, compared to genes that share either temporal or spatial co-expression.

13. Lines 347-349. Johnston et al. write, “Strikingly, 19 out of the 25 identified embryonic MEK/ERK downstream targets (Table S5) are downregulated by U0126 treatments during regeneration (Fig. 6Bi)”. Why is it striking? I would rather say, “as expected”.

Taking into account the reviewer’s comment and the new data, we have removed the word “strikingly” from the sentence. The reason we used “strikingly” was that considering the different context between embryonic development and regeneration, we found this observation striking and not necessarily expected. Additional data, particularly the observation that inhibiting cWnt signaling during regeneration does not affect embryonic cWnt downstream targets (Fig.8), confirm our idea that conservation of 'pathway-specific' downstream targets is not necessarily expected.

14. Lines 370-372. I do not doubt the involvement of Wnt and MEK/ERK signaling in the regulation of regeneration but I do not see any experimental confirmation of the placement of Wnt downstream of MEK/ERK. This study lacks systematic testing of what happens to regeneration upon upregulation and downregulation of Wnt as well as upregulation and downregulation of MEK/ERK as well as data showing which of the two (or maybe both) pathways is activated by the early wave of apoptosis. Please also do not forget to refer to the Hydra paper of Chera et al., 2009 when talking about the effect of apoptosis on Wnt.

Our additional experiments for the revised version of the manuscript should satisfy the reviewers’ comments. The Chera et al. paper was cited and discussed in the initial version of the manuscript (l. 441, l.449) and in the revised version in l.624, l.676, l.691.

15. Line 396. Remove “and”

Done.

16. Line 427. Why does the logic have to be reshuffled? What is the basis for this prediction?

See reply to the 2nd major comment above.

17. Line 429. Remove “and” from “other and vertebrates”.

Done.

18. Lines 460-463. Johnston et al. write that “cWNT and MEK/ERK activate a distinct set of downstream targets, while during regeneration MEK/ERK signalling is able to activate not only embryonic MEK/ERK targets but also embryonic cWNT as well as “regeneration-specific” genes.” This is a “false dichotomy”. There is a clear overlap between cWNT and MEK/ERK targets in the embryo (WntA, Sp8/9 – see Kraus et al., 2016 and Lebedeva et al., 2017). Please correct.

We’re unclear, or even in disagreement with the reviewers’ interpretation from the cited References in regard to our study.

In Kraus et al. 2016, the study nicely describes a pre-bilaterian blastoporal axial organizer. We could not find any sign of the gene SP8/9, any inhibition of MEK/ERK, or any other experiment that would suggest a control of *wntA* expression by MEK/ERK. The Kraus et al. 2016 study mainly used the cWnt activator 1-Azakenpaullone, and showed that among the tested genes, *wntA* can respond to 1-Azakenpaullone at the gastrula stage. These data extended previous observations from Röttinger et al, 2012 to later stages (blastula *versus* gastrula, respectively).

It is important to note here that in Kraus et al. 2016, the experiments correspond to global overactivation of the cWnt pathway that only show that *wntA* can respond to this global overactivation. Meaning that this is no evidence for *wntA* being a downstream target of cWnt signaling in normal developmental conditions. In fact, when performing Knock-Down experiments using a morpholino targeting TCF, the effector of cWnt signaling, *wntA* expression seems not affected, suggesting that *wntA* is not a target of cWNT signalling in this early developmental context (Fig. 10 in Röttinger et al, 2012). In line with these data, injection of a morpholino against β -catenin does not block *wntA* expression neither, but instead, extend its expression territory to more aboral tissues, suggesting that cWnt signaling is involved in restricting expression of *wntA* to the oral central ring domain (Leclere et al. 2016). Taken together, these data indicate that during embryogenesis *wntA* is NOT a downstream target of cWnt signaling at the oral pole under physiological developmental conditions at the late blastula stage in *Nematostella*.

On the other hand, *wntA* expression was downregulated under U0126 (MEK inhibitor) conditions at the late blastula (Layden et al, 2016; Additional File 8 – Table S8). Although, the injection of a morpholino targeting *erg*, a potential effector of MEK/ERK signaling during embryonic development of *Nematostella* (Amiel et al, 2017), did not yield in a significant downregulation of *wntA* expression. These data indicate that *wntA* is an Erg-independent MEK/ERK downstream target.

The gene *sp6/8* has also been identified in the microarray performed following U0126 treatments in late blastula stages (Layden et al, 2016; Additional File 8 – Table S8). Like *wntA*, *sp6/8* is not downregulated following the injection of morpholinos targeting *erg* (Amiel et al, 2017) showing that *sp6/8* is also an Erg-independent MEK/ERK downstream target.

Concerning the second reference indicated by the reviewer, Lebedeva et al., 2017, we could not find any publication corresponding to Lebedeva et al, 2017. In fact, the Google Scholar from Tatiana Lebedeva (which we believe is the author the reviewer is referring to) indicates only one paper for 2017, however, under the name of Tatiana Bagaeva. This paper addressed a *Nematostella* un-related topic.

The closest publication regarding the reviewers comment we could find and in which Tatiana Lebedeva is 1st author corresponds to Lebedeva et al, 2021 (DOI: 10.1038/s41467-021-24346-8). This publication includes a *sp* gene, but it is called *sp6-9*, and not *sp8/9* (which corresponds to the reviewer's concern). Although the spatial expression patterns are similar between the gene *sp6/9* (Lebedeva et al, 2021) and the gene *sp6/8* (from Layden et al 2016 and Amiel et al, 2017, which could be the gene that has been named *sp6/9* by Lebedeva et al, 2021), we could not find any sequence related to *sp6-9* to confirm this similarity.

In Lebedeva et al. 2021, experiments have been carried out at the gastrula stage and not at the blastula stage like in Röttinger et al 2012; Layden et al 2016 or Amiel et al 2017. Lebedeva et al. 2021 has identified *sp6-9* in a transcriptomic analysis following 1-Azakenpaullone treatments (meaning in a cWnt overactivation context) showing that *sp6-9* can respond to global cWnt overactivation. In another set of experiments *sp6-9* expression has been assessed by *in situ hybridization* following Knock Down of genes of interests (*bra*, *Imx*, *foxA* or *foxB*) or/and 1-Azakenpaullone treatments. These experiments shows that *sp6-9* expression is shifted orally when the cWnt downstream targets *bra*, *Imx*, *foxA* or *foxB* (all four expressed in the central ring) are functionally impaired. These results highlight the importance of these four genes (and cWnt signalling) for spatial restriction, confining *sp6-9* expression to the external ring and thus patterning the O-A axis during gastrulation. The 1-Azakenpaullone treatments under these KD conditions further confirm that *sp6-9* expression can be activated in global cWnt overexpression conditions, independently to the absence of one of those genes. In sum, we could not find any results that suggest that *sp6-9* expression is controlled by cWnt under physiological developmental conditions at the late blastula stage in *Nematostella*.

Taken together, we do not see how the mentioned publications from the reviewer show that the data we used for the comparison leading to our conclusions is a “false dichotomy”. As mentioned above we have slightly tempered our statement to take into account the genes, for which we have found evidence that they are controlled by MEK/ERK and cWNT signaling during embryonic development.

19. Line 676. Probably the title of the figure should be “ERK phosphorylation is required for regeneration”.

We agree with the reviewer and changed the title.

20. Figure 7A. The authors call this a network module, but it is just a list of MEK/ERK target genes. It will become a network module once the regulatory relationships between the targets are experimentally established.

We agree and have removed that sentence from the legend.

21. Figure 7B. This scheme is very suggestive but I do not think there is experimental evidence for it. Has it been shown that b-catenin signaling is required for MEK/ERK signaling in the embryos? I do not think there is a publication showing that b-catenin knockdown leads to the absence of MEK/ERK signaling in the embryos. Similarly, this study did not demonstrate experimentally that MEK/ERK is upstream of Wnt/b-catenin. If this is true, b-catenin target genes should be suppressed upon U0126 treatment, but this has to be checked first.

We agree with the reviewer. It was meant to indicate the temporal relationship rather than functional relationship. The new version of the figure (now Fig. 9) now focuses on the temporal deployment of the studied pathways rather than the functional relationships.

Reviewer #2 (Remarks to the Author):

The manuscript by Warner et al uses a re-analysis of prior transcriptomic data to address the similarities and differences between embryogenesis and regeneration in the model sea anemone *Nematostella vectensis*. The fundamental question of whether regeneration is simply a redeployment of development has been extensively posed in the literature in prior work. Hence the novelty of this study is primarily restricted to the organismal system and the systems biology methods employed to identify differentially expressed genes. As it appears this part of the work largely draws on analysis of published data, the key contribution amounts to the validation and functional experiments and therefore the impact of the work is fairly field-specific. In that regard, this manuscript provides an interesting starting point but falls significantly below the level of general insight expected for the journal. The experimental validation sections of the paper, which address roles for apoptosis and MEK/ERK signaling in regeneration are a big strength of the paper and represent an important advance in the field.

We thank the reviewer for the comments and hope that the revised version of the manuscript provides more detail and experimental evidence to support our claims. Previous studies that have addressed the question were and are presented in the manuscript and we strongly believe that our study goes further than what has been described so far. To address the reviewer's comments, we have conducted a significant number of additional experiments and functional analyses for this revised manuscript, which we believe will further increase the interest within the broader scientific community.

On the whole, the figures presenting the transcriptomic analysis are significantly below journal standards and the data do not clearly support conclusions drawn by the authors. Several highly general statements e.g. "This indicates the transcriptional dynamics of embryogenesis are more profound than those of regeneration" are so vague as to be meaningless as the analysis of bulk RNA-seq data likely obscures the transcriptional dynamics of critical cells at the wound site that contribute to regeneration.

We have added a sentence to the results section to place this sentence better into the comparative context of the manuscript. As the amputation we performed were below the pharynx (~1/3-1/2 of the body of juveniles), the program that is deployed during regeneration must allow the reformation of the main parts and structures of the animal (i.e., body-wall epithelia, mesenteries, pharynx, mouth, tentacles as well as the associated muscles and neural networks).

As the reviewer noted, the series of transcriptomics analyses presented in Fig 1 and 2 (and associated supplemental figures S1, S2, S3) were the starting point for a series of experiments related to the global question we addressed. They are a crucial aspect of our question we're addressing, *i.e.*, to what extent regeneration re-deploys embryonic gene networks. Thus, we decided to keep that statement as it relates not only to Fig. 1D but also to Fig. 1E and Fig. 3 leading to the global conclusion that regeneration is largely a partial re-use of embryonic development.

In this regard, a much higher resolution understanding could be gained by scRNA-seq or spatial transcriptomic approach. It is not clear whether or how this part of the work will have impact or utility to the broader community beyond serving as a starting point for the subsequent experiments.

The transcriptomic data we used and completed for our study (Warner et al., 2018) is already used by the community for various studies covering development and regeneration in a variety of research models (80 citations as of today). For the current study, we have chosen to use these bulk RNAseq data on purpose as we wanted to address the question on the relationship between embryonic development and regeneration as large as possible and as unbiased as possible, considering also the tissular aspects (i.e. the importance of the mesenteries to induce regeneration – Amiel et al – BioRxiv). Doing so we were able to span a large set of time points during embryonic development (22 time points) and regeneration (16 time points) enabling us to perform an extensive comparison between these two developmental trajectories. The outcome of the first part serves indeed as the starting point for the subsequent experiments of the current manuscript, but also open various venues for additional studies within the lab and the broader community.

We agree with the reviewer that the cell specific dynamics and regulatory sc- or spatial RNAseq would yield interesting complementary data. While we consider that these approaches will address other more specific questions (ex the comparison of cell fate trajectories), it is currently out of the scope for the present study and part of ongoing work in the laboratory.

The ZVAD experiments to inhibit apoptosis reveal an interesting requirement for apoptotic cell death in the initiation of regeneration and reflect one of the stronger logical components of the paper. It would be interesting to add an analysis of cell death following wounding without amputation to determine whether cell death can be seen as a consequence of wounding or reflects a component of the regenerative program proper.

In both conditions, puncture/wounding (non-regenerative – Dubuc et al. 2014) or amputation (regenerative – our study), the first wave of apoptosis is a consequence of wounding. The results from Dubuc et al. 2014 were and are still mentioned in our manuscript. However, the second and third waves of apoptosis we have identified and that are not located at the wound site anymore, seem to reflect a component of the regenerative program.

The TUNEL analysis suggests there may be wound-site specific and systemic cell death upon amputation and the authors could significantly expand the impact of the work by digging deeper in this area.

We agree with the reviewer that this is an interesting line of research and have performed additional experiments to gain a better understanding of the different waves of apoptosis (Fig.S9, briefly described above). We also highlight and discuss that the first wave of apoptosis is caspase-dependent but ERK-independent, while the 2nd wave of apoptosis seems caspase-independent, while ERK-dependent.

Similar to the analysis of cell death, the results on MEK/ERK could provide key new insights into *Nematostella* regeneration. Combined with the work on apoptosis, this is the strongest part of the paper. However, it remains unclear how direct or indirect the activation of downstream targets may be and how specifically U0126 treatment affects MEK/ERK signaling. Given the proliferation of genetic methodologies for *Nematostella*, are there not more rigorous approaches available to demonstrate specific gene requirements by CRISPR or shRNA knockdown?

We agree with the reviewer on the various points that were raised. ATACseq to gain insight into the regulatory elements and determine direct or indirect interactions during regeneration is part of another ongoing study from the laboratory and we decided to keep this aspect out of the present manuscript. We did this also for the sake of maintaining the same level of comparison between embryonic development and regeneration. As for the use of other functional approaches, unfortunately, we and the community are not quite there yet. Currently, there are no available conditional KO approaches, and our attempts to soak morpholino or siRNA in juveniles did not produce exploitable results.

Reviewer #3 (Remarks to the Author):

The manuscript by Johnston et al. uses a comparative approach to determine the relationship between embryonic development and regeneration at both molecular and cellular levels. While defining the key factors that distinguish regeneration from development is fundamentally important, experimentally, only a few research organisms offer the possibility to address this question. In this context, the authors leveraged the accessibility of embryonic development and regeneration in the sea anemone *Nematostella vectensis* and have generated and analyzed impressive transcriptomic data covering the temporal dynamics of both embryogenesis and regeneration. Building on this comparative approach and functional data involving ERK signaling, the authors identified and validated gene expression patterns that reflect a rewiring of the embryonic genetic program. At the cellular level, apoptosis appears to be required for regeneration, which emphasizes the role of this cellular process as a core mechanism deployed across different regenerative species. The great strength of this study is that the authors have produced significant transcriptomic resources for the scientific community, and the advancements presented here certainly establish this sea anemone as an attractive research organism to study the developmental origins of regeneration. In general, the experiments are well performed throughout the manuscript. However, the interpretations of some results would require clarification or additional controls to address the following concerns:

Major concerns:

1- The authors claim that “Regeneration is transcriptionally modest compared to embryonic development”. However, this conclusion is based on comparing oral pole regeneration, which does not represent the full regenerative capacity of the polyp while embryogenesis builds a whole organism. The authors should reconsider this statement as the current data do not support it.

We thank the reviewer to point this out and have added “oral” and replaced “is” by “appears” in this sentence and included the raised comment in the manuscript. Rather than a contradiction, we consider the experimental design for the comparison we carried out (embryonic development vs oral regeneration) an explanation for the observed difference in transcriptional dynamics. We have included this in the updated discussion that was also extended to explain possible differences with other studies.

The body parts following amputation that are used for the transcriptomic analysis cover the reformation of a large part of the body. In fact, the sub-pharyngeal amputation we performed on the juveniles, measuring ~500µm in length, are nearly corresponding to mid-body amputation. This means that the molecular program deployed during regeneration must allow the reformation of most of the body of the animal (i.e., body-wall epithelia, mesenteries, pharynx, mouth, tentacles as well as the associated muscles and neural networks).

Regeneration in *Nematostella* occurs only when mesenterial tissues are present (Amiel et al, BioRxiv), making it impossible to launch regeneration from a mesentery-less physa. To cover the largest possible regenerative program, we have decided to perform the transcriptome on entire regenerating polyps and not only the amputation site. Furthermore, systemic gene

expression changes in tissues distant from the site of injury has been documented in numerous model systems, including in *Nematostella* as shown in a recent study (Cheung et al, 2025). Expression patterns induced in the mesenteries after amputation (ex. Fig. 8C) are in line with this idea. Thus, our data and results consider any patterning and scaling event even far from the amputation site and making our intra-species comparison as broad and as unbiased as possible.

2- An experimental validation of ‘regeneration specific’ genes is missing. In this context, an attractive category is the list of genes that are taxonomically restricted. This work has the potential to inform us about their integration in the gene regulatory logic. For instance, Is there any bias in their spatio-temporal deployment during regeneration?

A first, global, validation of the regeneration-specific genes set, that include the pro-apoptotic gene *bax*, is the assessment of the role of apoptosis. This revealed that compared to embryonic development (and taking into account the reference mentioned by the reviewer below), apoptosis seems to be a regeneration-specific process. As the focus of the present study is to understand to what extent regeneration redeploys embryonic elements, we have included *in situ* hybridizations for regeneration-specific genes in Fig. 3 (screen of embryonic MEK/ERK or cWnt downstream targets as well as the regeneration-specific genes up-regulated at 20hpa) and 8 (regeneration-specific genes down-regulated by U0126 treatments). An in-depth functional assessment of the role of these genes during regeneration is part of an ongoing project in the laboratory.

Considering the reviewer’s comment and to provide additional insight into the expression of the regeneration specific genes, we have added a table indicating the regeneration expression clusters these genes belong to (Fig. 1F), highlighting an increased number of regeneration specific genes that are part of the expression cluster R9 (36 genes) and R6 (29 genes), followed by R2 (14 genes) and R3 (13 genes).

More globally, the manuscript, presents spatial expression patterns for 5 regeneration specific genes: *bax*, *wls*, *carm*, *cad* & *egln* (Fig. 3 & Fig. S4; Fig. 4, Fig. 8). All genes belong to expression cluster R6, however, their expression pattern is not identical within the various tissues at the amputation site (i.e., body wall epithelia vs mesenteries), suggesting complementary regulatory elements that drive their expression following amputation. Deciphering this in detail as well as the role of these genes during the process of regeneration is out of scope for the current manuscript and is currently being investigated in another project from the lab.

3- The authors also mentioned that “Apoptosis is specifically required for regeneration” without providing strong evidence to support this claim. Many studies show embryos or larvae might look morphologically normal while cell differentiation can be dramatically impacted when using molecular profiling. In addition, a peak of cell death has been reported during late larval development (Zang and Nakanishi, 2020). An obvious experiment would be to treat late larvae with the pan-caspase inhibitor Z-VAD and test whether development proceeds normally.

We thank the reviewer for pointing out the publication from Zang & Nakanishi that we have missed. This reference is now cited in our manuscript and the statement adjusted accordingly.

We have now included experiments (Figure 4A) that show not only that embryonic development occurs normally, but also that metamorphosis occurs following Z-VAD treatments (0-polyp & planula-polyp) at the same rate than DMSO treated embryos.

4- It is exciting to see the involvement of apoptosis in *Nematostella* regeneration. Could the authors explore the transcriptomic data and show the expression of developmental signals that are provided by these dying cells.

We agree that this is an interesting result. Although the more in-depth analysis of the developmental signals provided by apoptotic cells appear to be out of the scope of the present paper, we performed additional transcriptome for the current study including under apoptosis inhibition during regeneration. The transcriptomic data following Z-VAD treatments indicates a series of developmental genes such as "*FoxL, Nidogen, Neuropilin, Bmp1, etc.*" (Fig. 7D, Table SX). While from the current data it seems unlikely that Wnt's, like in Hydra or Zebrafish, are signals emitted by apoptotic cells, our results could indicate that apoptosis participates in modulating the extracellular matrix and thus the associated signaling.

Minor concerns:

- The labeling of the panels with multiple letters is not simple to follow and creates confusion.

We have revisited the Figure labelling and hope that the new version facilitates the lecture.

- There is a typo in "Apoptosis is a and regeneration-specific process."

Done.

Reviewer #1 (Remarks to the Author):

The authors have significantly improved the paper, and experimentally it is now a very solid work. There are several specific things, which, in my opinion, need to be done for the manuscript to become publishable (see the point-by-point below). I also suggest that the authors shorten the discussion making it less speculative and focusing it on the implications of their main findings rather than on what other questions would be potentially interesting to address.

We thank the reviewer for highlighting the improvements of our revised manuscript as well as for the additional constructive point-by-point comments / recommendations that we addressed below and in the manuscript. We have also revisited and shortened the discussion.

Point-by-point comments in order of appearance in the text (not in the order of importance!).

Line 163. “Regeneration-specific” genes may be regeneration-specific only in this particular comparison. For example, *wntless* is a maternal transcript abundantly present throughout development in the pharyngeal and body wall ectoderm.

We agree that *wntless* is not solely expressed during regeneration but also during embryonic development. In fact, as stated on l. 160-161 from the 123 “regeneration-specific” genes, “*.78 genes are expressed during embryonic development but are not considered dynamic from our differential expression analysis above (Fig.1F)*”. *wntless* is among these 78 genes.

As stated in the text (see l.154-157) and in our previous reply to a similar comment, we defined the “regeneration-specific” genes not because they are only expressed during regeneration, but that their transcriptional expression is dynamic during regeneration ($fC > 2$ -fold) and not during embryonic development ($fC < 2$ -fold).

As stated in l.158-159, from the 123 “regeneration-specific” genes highlighted in this analysis, only “*45 are detectable specifically during the regeneration process indicating they are transcriptionally silent during embryogenesis*”. We have replaced “specifically” by “exclusively” to clarify this point.

Although we consider that we have clearly defined the term “regeneration-specific”, we remain cautiously open to replacing this concise term into a longer description. Specifically, the term “regeneration-specific” could be replaced by “genes with a regeneration-specific expression dynamics”. But as we use the term frequently throughout the manuscript, this would make it more heavy to read and extend the manuscript length.

Line 333. Fig. S8: In contrast to the staining on Fig. 4L, where nuclear b-cat-mCherry is certainly real at 12 and 24 hpa, the authors misinterpret the b-cat-mCherry staining on Fig. S8D (by the way, D is present in the legend but missing on the figure!). The

small circles in the middle of the cells, which the arrowheads are pointing at are not nuclei. They are crowns of microvilli surrounding the cilium, - a similar picture is shown in Fig. 1 of the Momose et al., 2012 on *Clytia*, and on your own Fig 4L in the 48 hpa image. Nuclei are much bigger - see DAPI.

We thank the reviewer for pointing this out and for clarification, we have revisited our images and accordingly replaced the images in Figure S8D. We also added the missing “D” in this Figure.

Also, the authors show significant depletion of the mCherry signal in the cell contacts in iCRT14 in comparison with DMSO, which should not happen, since iCRT14 prevents b-cat/Tcf interaction, not something else.

Indeed, the mCherry staining in the membrane seems also affected, however, we would not claim it to be significant as we did not perform any quantification. One explanation of this observation might be connected to the RNAseq data we obtained following iCRT14 treatments that revealed that ECM associated genes were downregulated under those conditions (Fig. 7C). Thus, rather than a direct effect of iCRT14 on the membrane pool of β -catenin, the reduced mCherry staining at the membrane after iCRT14 treatments might be indirect and associated with reduced ECM associated genes, involved in cell-cell interactions.

Another point: I am not sure what “a”, “b”, “abc”, “ac” and “d” mean on top of the error bars on (C), but to me this plot shows that AZ is the only chemical with a clear effect on b-cat signaling. iCRT14 does not have a significant effect on b-cat in comparison to DMSO. It is not clear to me, how iCRT14 can abrogate the effect of AZ if it does not have an effect on topflash-driven expression when administered alone. A good control for the functionality of iCRT14 would be qPCR or RNA-Seq on b-cat target genes during regeneration. It should be reciprocal in iCRT14 and AZ - at least for genes upregulated in AZ.

The letters (“a”, “b”, “abc”, “ac”, “d”) above the error bars on (C) indicate groups that are statistically different based on the post-hoc test, as detailed in Supplementary Table 26. We added this information also in the legend of the corresponding figure. “Different letters above the bars indicate statistically significant differences among treatments ($p < 0.05$) according to two-sided pairwise Kruskal–Wallis tests followed by post hoc Dunn tests with Benjamini–Hochberg adjustment. Statistical analyses are provided in Sup Table 26”

We don't really grasp the rationale why iCRT14 can't significantly abrogate the overactivation of β -catenin (measured by luciferase activity) in AZ treated animals when treated together. This is what our data show.

However, we agree that the observation that iCRT14 doesn't visibly affect topflash activity under physiological conditions might be somewhat puzzling. We propose that the absence of significant reduction in this context might be explained by the faint/basal level of luciferase activity in DMSO treated controls that are not sufficient to detect any decrease using this technique under physiological condition. In line with this idea, AZ-treated embryos co-treated with iCRT14 reach the same levels of luciferase activity as DMSO or iCRT14 alone.

Although previous studies (Marlow et al, 2013; Faltine-Gonzalez et al, 2023) that we cite in the manuscript have used iCRT14 in *Nematostella* to inhibit the cWnt pathway, we developed experimental procedures to visualize and show the effect of this pharmacological inhibitor on cWNT signaling either using Topflash-driven luciferase expression (Fig. S8C) and on endogenous β -catenin localization directly (Fig.S8D) as well as on regeneration (Fig. 5,6,7). In summary:

1 - iCRT14 prevents the effects of AZ induced luciferase expression

2 - regeneration-induced nuclearization of the endogenous β -catenin at the amputation site is prevented in iCRT14-treated polyps

3 - The iCRT14 caused phenotype (i.e., block of the regeneration process at “stage 2”) is in line with the activation pattern of the endogenous β -catenin starting between 12-20hpa.

All together, our complementary results are in line with an inhibitory effect of iCRT14 on endogenous β -catenin / cWnt signaling. In our study, we have performed RNAseq on iCRT14 treated animals during regeneration (See Fig. 7). In our opinion, regarding the main question of our study, all our results do not warrant further RT-qPCR or RNAseq experiments on AZ treated animals during regeneration.

Lines 334-335: Clumsy sentence. MEK/ERK and cWnt do not block development and patterning. Experimental modulation of MEK/ERK and cWnt does.

We thank the reviewer for pointing this out. The sentence has been changed to:

"Previous studies have shown that perturbation of MEK/ERK and cWnt perturbs embryonic development and patterning in N. vectensis."

Lines 352-353: The authors write “iCRT14 treated polyps did not display β -catenin nuclearization at 24hpa (Fig.S8C)” – but their controls also do not show b-cat nuclearization - at least on Fig. S8. See comment to Line 333. There is clear nuclear b-cat-mCherry signal on Fig. 4L at 12 and 24hpa.

This comment has been addressed in a previous comment above. Following the reviewer’s advice we have re-assessed our images and replaced the ones from Fig. S8D that now shows the abrogation of endogenous β -catenin nuclearisation following iCRT14 treatments, compared to DMSO control.

Line 363: Replace “as” with “in comparison to”

Done.

Line 445: On Fig. 7C, the GO-terms are completely unreadable without a 5x zoom, and certainly not on the printout. If this is a main text figure, every part of it has to be readable on a standard page without zooming.

We have taken into account the reviewers comment and now provide a summary Figure for 7C (main figure) and have moved the GO terms to Figure S14 (supplementary figure) so that they are bigger and more readable.

Lines 451-461: How do the authors explain that MAPK signaling is upstream of Wnt, but the genes controlled by Wnt are not affected by MAPK inhibition?

The reviewer raises a good point that we have attempted to answer in our revised version (l.462-469). While the temporal aspect we stressed might be an explanation, we now propose two additional possible reasons that might be even more compelling: a) the presence of compensatory regulatory mechanisms that drive cWnt target genes during regeneration in *N. vectensis* independently of MEK/ERK signaling, or b) spatially distinct actions of MEK/ERK signaling that affect amputation-induced activation of cWNT signaling in one region (e.g., the amputation site BWE), while cWNT signaling activation can remain unaffected in another region (e.g., mesenteries). We completed this paragraph in the latest revision with the above mentioned elements.

Lines 476-481: The authors write “Systematically comparing these pools of genes (emb_cWnt, emb_ERK, regen_spe) with the up- or down-regulated genes following inhibition of MEK/ERK (Tables S20,S21), apoptosis (Tables S22,S23), or cWnt (Tables S24,S25) during regeneration, we observed that MEK/ERK signaling during regeneration has the most widely extended control on gene expression for all three pools (Fig.8; Fig.S15).” This makes total sense, but doesn't it contradict what they wrote on lines 451-461 (that target lists barely overlap)?

The two analyses mentioned by the reviewer are addressing different questions: (i) downstream targets of MEK/ERK, apoptosis or cWNT during regeneration (Figure 7), (ii) to what extent are genes that are either downstream targets of MEK/ERK or cWNT signaling during embryonic development, or that belong to the set of “regeneration-specific” genes, are controlled by either MEK/ERK, apoptosis or cWNT during regeneration (Figure 8). The results therefore provide complementary, not contradictory information.

More specifically, the statement about the downstream targets that barely overlap (l.451-461) makes reference to Fig. 7, *i.e.*, the downstream targets whose expression are affected by U0126, Z-VAD or iCRT14 treatments during regeneration.

Fig.8 compares the downstream targets identified in Fig.7 to (a) embryonic downstream targets of MEK/ERK signaling, (b) downstream targets of cWnt signaling and (c) the “regeneration-specific” genes that are up-regulated at 20hpa. These three pools of genes make reference to Fig. 3.

The latter comparison highlights that part of the embryonic MEK/ERK or embryonic cWnt downstream targets as well as some of the “regeneration-specific” targets are controlled by MEK/ERK signaling during regeneration and not by cWNT or apoptosis.

To clarify this point, we have added the reference to Fig. 7 and Fig. 3 to the manuscript.

Lines 495-502: I do not find labels like Bi, Bii, and Biii on Fig. 8. I also do not see any insets on Fig. 7B.

The reviewer is correct and as the panels of Fig. 8 contain specific subtitles, we have removed the indications “i,ii and iii” in the manuscript. The insets are in Fig. 8B and we fixed this in the manuscript.

Line 514: This is not a GRN. Unlike anything published by Eric Davidson, who was using the same type of graphics for GRN representation, here, it is just a list of downstream targets of MEK/ERK without any experimentally proven or even hypothetical regulatory links between them. It occupies figure space without providing any information a supplementary table cannot provide. Please remove.

We agree with the reviewer that the GRN blueprint we established, shown in Fig. 8D, does not include any regulatory interactions between the MEK/ERK downstream targets. We use the term GRN blueprint (Röttinger et al, 2012; Amiel et al, 2017) to indicate putative transcriptional interactions and clearly state in the Figure legend that no assumption on direct or indirect targets are made.

L.547-548 further states: “This GRN blueprint indicates the potential direct or indirect downstream targets of MEK/ERK signaling at the onset of regeneration”. We have changed the term “GRN” by “GRN blueprint” in this sentence. We decided to keep this diagram / GRN blueprint as it a) not only provides a clear visual on the MEK/ERK downstream target genes during regeneration but b) also includes spatial information of these downstream targets. This GRN blueprint diagram might be of usage to the community for developing and testing regulatory hypotheses in follow-up studies.

Line 534-536: The authors write that their results “support a model in which regeneration reuses core developmental programs such as cell type specification and patterning, while also activating unique, regeneration-specific regulatory elements and transcription factors”. They may be right in principle, but their results do not show anything of this sort. Out of 123 “regeneration-specific genes” in Table S1, there is a single transcription factor, Msx2-like, however, single-cell transcriptomics atlas of Cole et al., 2024 shows that it is strongly expressed in the retractor muscle and also in one particular neuroglandular cell type in a normal adult polyp. So, it may be “regeneration-specific” only in the comparison made by Andreoni-Pham et al., but not otherwise. Since there is no ATAC-Seq data in the manuscript, the authors cannot say that their data supports the existence of “regeneration-specific regulatory elements” either.

It seems that the way we formulated the sentence cited by the reviewer might lead to confusion in regard to what we defined as “regeneration-specific” genes *versus* the “regeneration-specific regulatory elements and transcription factors”. Only the latter is relevant to the cited sentence above.

To clarify this point, the revised sentence (changes are in blue) now reads: “Globally, our results support a model in which regeneration reuses core developmental programs such as cell type specification, **proliferation and** patterning,

and suggest that regeneration activates gene expression via regeneration-specific regulatory elements”.

The rationale for this sentence is the following. Fig. 1, Fig. 2 and associated Supplementary Figures, show that there is a large overlap of genes dynamically expressed between embryogenesis and regeneration, and that some of these genes have been associated with cell type specification, patterning & proliferation.

In addition, Fig. 2G,H highlights for example the expression dynamics of the transcription factors *pcf* and *runx*. During embryonic development, their temporal expression patterns are different and they belong to different embryonic expression clusters (EX and EX), suggesting that they are regulated by different elements. Interestingly, these two TFs are co-expressed during regeneration (i.e., they share a highly similar temporal expression profile and belong to the same regeneration expression cluster R6), suggesting that they are regulated by the same regeneration-specific elements.

Line 657: wnt7

Done

Line 703: The scheme on Fig. 9 is incorrect. MEK/ERK is responsible for the initiation of the endomesodermal expression in the plate in the 6-8 hpf blastula stage (Haillot et al., 2025); that is the same time when maternal beta-catenin goes into the nuclei of all future ectodermal cells (Lebedeva et al., 2025). So, there has to be not only the orange but also the green arrows to each embryo cartoon.

We thank the reviewer for pointing this out. However, rather than incorrect, we agree that the scheme on Fig.9 was missing an element. Following the reference cited by the reviewer we have added a green arrow to the blastula stage (no indication of MEK/ERK activity at the cleavage stage is indicated in Halliot et al, 2025). We also added a green arrow to the post-metamorphic juveniles stage as MEK/ERK signaling is required for metamorphosis (Rentzsch et al, 2008).

Reviewer #2 (Remarks to the Author):

The authors have incorporated significant new functional experiments into the manuscript in F5 and F6. This addresses one major concern. Still, their rebuttal to my comments offered mostly argumentation and minimal specific changes in response to the original comments (excepting one changed sentence and the analysis of the waves of apoptosis in SF9). On the whole, the paper is improved and scientifically sound but I maintain some concerns about the generality of the bulk RNA-seq analysis, the lack of single cell resolution, and thus the general impact of the work relative to the current literature.

We thank the reviewer for his comments and to recognize the improvement of the manuscript as well as the significant amount of functional data that were implemented in this updated version of our manuscript.

As mentioned in our previous response, we agree with the reviewer that the cell specific dynamics, regulatory, sc- or spatial RNAseq would yield interesting complementary data. While our current scope was to gain a broad understanding of the redeployment of embryonic processes during regeneration, we consider that the single-cell approaches will address other more specific questions (e.g., the comparison of cell fate trajectories) that are part of ongoing work in the laboratory and that we're looking forward to sharing with the community in the future.

Reviewer #3 (Remarks to the Author):

The revised manuscript by Andreoni-Pham, Johnston & Warner et al. represents a substantial improvement over the initial submission. The authors have effectively addressed the major concerns by incorporating additional experiments that strengthen the connections between their transcriptomic analyses and claims. The manuscript now presents a more cohesive narrative that integrates descriptive and functional approaches. I believe the work is now suitable for publication in Nature Communications.

We thank the reviewer for his comments and recognition of the substantial improvement of our current manuscript. We are glad to see that we convincingly addressed the previous concerns.